# Few-shot Generation via Recalling Brain-Inspired Episodic-Semantic Memory

**Zhibin Duan,**[*] **Zhiyi Lv,**[*] **Chaojie Wang,**[†] **Bo Chen,**
National Laboratory of Radar Signal Processing
Xidian University, Xi'an, Shaanxi 710071, China
`xd_zhibin@163.com, xd_silly@163.com, bchen@mail.xidian.edu.cn`

**Bo An**
Nanyang Technological University

**Mingyuan Zhou**
The University of Texas at Austin

## Abstract

Aimed at adapting a generative model to a novel generation task with only a few given data samples, the capability of few-shot generation is crucial for many real-world applications with limited data, *e.g.*, artistic domains. Instead of training from scratch, recent works tend to leverage the prior knowledge stored in previous datasets, which is quite similar to the memory mechanism of human intelligence, but few of these works directly imitate the memory-recall mechanism that humans make good use of in accomplishing creative tasks, *e.g.*, painting and writing. Inspired by the memory mechanism of human brain, in this work, we carefully design a variational structured memory module (VSM), which can simultaneously store both episodic and semantic memories to assist existing generative models efficiently recall these memories during sample generation. Meanwhile, we introduce a bionic memory updating strategy for the conversion between episodic and semantic memories, which can also model the uncertainty during conversion. Then, we combine the developed VSM with various generative models under the Bayesian framework, and evaluate these memory-augmented generative models with few-shot generation tasks, demonstrating the effectiveness of our methods.

## 1 Introduction

A remarkable capability of human intelligence is its ability to quickly grasp the concepts of new objects that it has never encountered before [1]. Essentially, this rapid adaptation is achieved by utilizing the past memories stored in the human brain, which can greatly improve the efficiency of later learning [2, 3]. Unlike human brain, existing machine learning methods often require to be trained on a large amount of data when facing new tasks, motivating the research interests on few-shot learning that aim at efficiently solving these unseen tasks with only a few given data samples [4–6]. While there have been promising processes for few-shot adaptation on classification tasks, less work has been done on few-shot generation [7–10], which is mainly due to the challenging nature of learning the generative process with only a few samples in an unsupervised manner [11, 12].

For existing deep neural networks, to improve their abilities of few-shot adaptation, one of the most effective methods is to introduce an augmented memory module, which can store the memory of past experiences to adapt the model to a new task quickly [6, 13–16]. However, the most widely used memory mechanism in exisiting works is short-term [17] with limited capacity, which focuses on the

---

[*]Equal Contribution
[†]Corresponding Author

37th Conference on Neural Information Processing Systems (NeurIPS 2023).

collection of personal experience related to a particular place or time. To remove the limitation of short-term memory, Zhen et al. [18] introduce the concept of semantic memory, which allows the storage of general conceptual information into meta-learning to acquire long-term knowledge for few-shot learning. Although these memory modules have separately achieved attractive progress in few-shot learning [1, 2, 19, 20], there is little work to explore a memory mechanism to organically combine the semantic and episodic memories, and further apply it on few-shot generation tasks.

Inspired by the memory mechanism of the human brain, where both episodic and semantic memory work together to help humans understand the world around us and also accomplish creative tasks, in this work, we first develop a variational structured memory module (VSM), which can simultaneously store both episodic and semantic memories to improve the few-shot generation capability of the generative models. Then, we design two important processes in VSM: 1) structured memory recall, which retrieves the relevant information of previous tasks from VSM to be applied on current task; 2) structured memory update, which continuously collects data information from training tasks and gradually consolidates the structured memories stored in VSM. Finally, we combine VSM with a branch of few-shot latent variable models, specifically neural statistician (NS) [7], and develop a series of memory-augmented latent variable models for few-shot generation.

The main contributions of this paper can be summarized as follows:

- Inspired by the memory mechanisms of the human brain, we develop a novel variational structured memory module (VSM) that can concurrently store both episodic and semantic memories, effectively aiding generative models in memory recall during sample generation.

- Meanwhile, we introduce a bionic memory updating strategy for the conversion between episodic and semantic memories stored in VSM, where episodic memory can provide detailed episodes that will be converted later and stored in semantic memory.

- Through incorporating VSM with NS, we develop a novel hierarchical memory-augmented neural statistician (HMNS), whose generative process can be specified with VAE or diffusion model, leading to VSM-VAE and VSM-Diffusion, respectively.

- Extensive experiments demonstrate that our developed VSM-VAE and VSM-Diffusion can achieve promising generative performance on few-shot generation tasks.

## 2   Related work

**Few-Shot Latent Variable Models:** For few-shot generation, a series of latent variable models based on NS [7] have been developed [9–12], which can be further improved by adopting a non-parametric formulation for the context [21], introducing a hierarchical learnable aggregation for the input set [11], increasing expressivity for the conditional prior [10], or exploring supervision [22]. Specifically, NS [7] introduces a context constant variable to specify the summary statistics of a dataset, which provides a shared prior for these samples within the same dataset, contributing to handling new datasets at test time; GMN [9] defines a variational model to learn a per-sample context-aware latent variable. Distinct from previous works, our method introduces a memory module to collect historical datasest information to inference the context of a new dataset.

**Memory-Augmented Neural Networks:** It has been shown that augmenting existing neural networks with an external memory module can improve their capability of few-shot learning [23]. Specifically, these memory-augmented neural networks mainly collect the information stored in the support set of the current task [6, 15, 16], which will be wiped form episode to episode [17] instead of maintaining long-term information that has been shown to be crucial for effectively learning new unseen tasks. For instance, variational semantic memory [18] is proposed to accumulate and store the semantic information from previous tasks for the inference of the prototypes of new tasks. Distinct from previous methods only constructing semantic memory for the prototype network, in this paper, we develop a structured memory module to store the historical information at both the episodic and semantic sides, which can largely enhance the model's capability of few-shot generation.

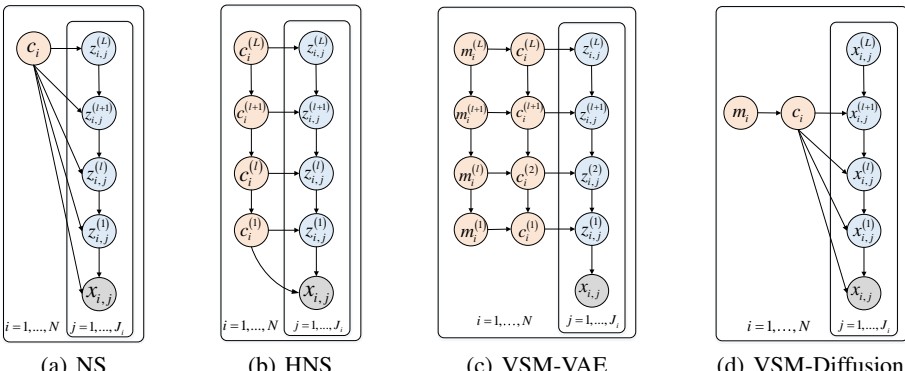

| (a) NS | (b) HNS | (c) VSM-VAE | (d) VSM-Diffusion |

Figure 1: The generative process of (a) neural statistician (NS), (b) hierarchical neural statistician (HNS), (c) memory-augmented variational autoencoder (VSM-VAE), and (d) memory-augmented difusion model (VSM-Diffusion), where $\boldsymbol{x}_{i,j}$ denotes the observed data sample, $\boldsymbol{z}_{i,j}^{(l)}$ is the latent variable at the $l$-th hidden layer, and $\boldsymbol{c}_i^{(l)}$ and $\boldsymbol{m}_i^{(l)}$ are the context variable and memory variable of dataset $D_i$, respectively. More details can be found in Section 3.2.

## 3 Memory-Augmented Neural Statistician

### 3.1 Preliminary

**Few-shot Generation**: Few-shot generation [22] borrows a similar setting of few-shot learning [4] but for a more challenging task to generate objects given only a few samples of that object at test time. Specifically, given a set of training datasets $D_{old} = \{D_i\}_{i=1}^{N}$, where each dataset $D_i = \{\boldsymbol{x}_{ij}\}_{j=1}^{J_i}$ consists of $J_i$ data samples, the goal of few-shot generation is to train a generative model $p_\theta(\boldsymbol{x})$ that is trained on given datasets $D_{old}$ but can be quickly adapted to a new unseen task given only a few data samples collected from a new dataset $D_{\text{new}}$.

### 3.2 Memory-Augmented Neural Statistician

Through incorporating memory mechanism with few-shot latent variable models, specifically NS [7] in Figure 1(a), in this paper, we develop a novel memory-augmented NS (MNS) for few-shot generation.

**Generative Process of MNS:** The design of MNS aims at introducing a memory module to store and update the data information at various semantic levels, which can be recalled when facing a new generative task. As shown in Figure 1(c), the generative process of hierarchical MNS is formulated as:

$$p_\theta(D_{old}) = \int\int\int \prod_{i=1}^{N}\prod_{j=1}^{J_i} p_\theta\left(\boldsymbol{x}_{i,j} \mid \boldsymbol{z}_{i,j}, \boldsymbol{c}_i, \boldsymbol{m}_i\right) \tag{1}$$

$$\prod_{i=1}^{N}\prod_{j=1}^{J_i} p_\theta(\boldsymbol{z}_{i,j} \mid \boldsymbol{c}_i) \prod_{i=1}^{N} p_\theta(\boldsymbol{c}_i \mid \boldsymbol{m}_i) \prod_{i=1}^{N} p(\boldsymbol{m}_i) d\boldsymbol{z}_{i,j}\, d\boldsymbol{c}_i\, d\boldsymbol{m}_i.$$

To clarify the meanings of $\boldsymbol{z}_{i,j}$, $\boldsymbol{c}_i$, and $\boldsymbol{m}_i$: $\boldsymbol{z}_{i,j}$ denotes the per-sample latent variables of the $j$th sample in dataset $D_i$, $\boldsymbol{c}_i$ denotes the task-specific context variable of dataset $D_i$, whose detailed definition can be found in [7], and $\boldsymbol{m}_i$ denotes the task-specific memory variable, which stores the data information of previous tasks that will be adapted to the new task.

### 3.3 Variants of Hierarchical Memory-Augmented Neural Statistician

Through specifying the generative process of HMNS and replacing its memory module with the variational structured memory (VSM) as described in Section 4, we can obtain a series of model

variants, such as VSM-HVAE and VSM-Diffusion, both of which can achieve promising generative performance on few-shot generation task.

### 3.3.1 VSM-VAE

**Generative Process of VSM-VAE:** To capture the hierarchical data information at various semantic levels, we extend the latent variables in Eq. (1) to a hierarchical version, specifically $z_{i,j} = \{z_{i,j}^{(l)}\}_{l=1}^{L}$, $c_{i,j} = \{c_{i,j}^{(l)}\}_{l=1}^{L}$, and $m_{i,j} = \{m_{i,j}^{(l)}\}_{l=1}^{L}$, which will be included in a variational autoencoder (VAE) with $L$ hidden layers, leading to a VSM-VAE shown in Figure 1(c). Under the generative framework of MNS, using the same symbol notation in Eq. (1), the per-sample generative process of $x_{i,j}$ in the dataset $D_i$ can be formulated as

$$p_\theta(x_{i,j}) = \int \int \int p_\theta(x_{i,j} \mid z_{i,j})p_\theta(z_{i,j} \mid c_i)p_\theta(c_i \mid m_i)p(m_i)\,dz_{i,j}\,dc_i\,dm_i, \tag{2}$$

where $p_\theta(z_{i,j} \mid c_i)$, $p_\theta(c_i \mid m_i)$, and $p(m_i)$ are formulated as the product of corresponding hierarchical latent variables as follows

$$p_\theta(z_{i,j} \mid c_i) = \prod_{l=1}^{L} p_\theta(z_{i,j}^{(l)} \mid z_{i,j}^{(l+1)}, c_i^{(l)}), \;\; p_\theta(c_i \mid m_i) = \prod_{l=1}^{L} p_\theta(c_i^{(l)} \mid c_i^{(l+1)}, m_i^{(l)}), \tag{3}$$

$$p_\theta(m_i) = \prod_{l=1}^{L-1} p_\theta(m_i^{(l)} \mid m_i^{(l+1)})p(m_i^{(L)}), \;\; p(m_i^{(L)}) = \mathcal{N}(0, I). \tag{4}$$

**Training Objective of VSM-VAE:** For VSM-VAE, the training objective is to maximize the evidence lower bound (ELBO) of a set of training datasets $\{D_i\}_{i=1}^{N}$, which can be expressed as

$$\mathcal{L}_{\text{ELBO}} = \sum_{i=1}^{N} \mathbb{E}_Q\left[\log p_\theta(D_i \mid c_i)\right] - \mathbb{E}_Q\left[\log \frac{q_\phi(c_i \mid D_i, m_i)}{p_\theta(c_i \mid m_i)}\right] - \mathbb{E}_Q\left[\log \frac{q_\phi(m_i \mid D_i, \mathbf{M})}{p_\theta(m_i)}\right]. \tag{5}$$

The first item in Eq. (5) can be further written as

$$\mathbb{E}_Q\left[\log p_\theta(D_i \mid -)\right] = \sum_{j=1}^{J_i} \mathbb{E}_Q\left[\log p_\theta(x_{i,j} \mid z_{i,j}^{(1)})\right] - \sum_{l=1}^{L} \mathbb{E}_Q\left[\log \frac{q_\phi(z_{i,j}^{(l)} \mid x_{i,j}, z_{i,j}^{(l+1)}, c_i^{(l)})}{p_\theta(z_{i,j}^{(l)} \mid c_i^{(l)}, z_{i,j}^{(l+1)})}\right]$$

$$\mathbb{E}_Q\left[\log \frac{q_\phi(c_i \mid -)}{p_\theta(c_i \mid m_i)}\right] = \sum_{l=1}^{L} \mathbb{E}_Q\left[\log \frac{q_\phi(c_i^{(l)} \mid D_i, c_i^{(l+1)}, m_i^{(l)})}{p_\theta(c_i^{(l)} \mid c_i^{(l+1)}, m_i^{(l)})}\right], \tag{6}$$

$$\mathbb{E}_Q\left[\log \frac{q_\phi(m_i \mid -)}{p_\theta(m_i)}\right] = \sum_{l=1}^{L} \mathbb{E}_Q\left[\log \frac{q_\phi(m_i^{(l)} \mid D_i, \mathbf{M})}{p_\theta(m_i^{(l)} \mid m_i^{(l+1)})}\right],$$

and $Q$ denotes the variational distribution defined by the inference network of VSM-VAE

$$Q = \prod_{l=1}^{L} q_\phi(z_{i,j}^{(l)} \mid x_{i,j}, z_{i,j}^{(l+1)}, c_i^{(l)})q_\phi(c_i^{(l)} \mid D_i, m_i^{(l)})q_\phi(m_i^{(l)} \mid D_i, \mathbf{M}), \tag{7}$$

where $\mathbf{M}$ indicates the memory module to store the semantic information collected from previous tasks, and will be introduced in Section 4.

### 3.3.2 VSM-Diffusion

**Generative Model of VSM-Diffusion:** Further, through specifying the generative process of HMNS with the diffusion model, we can obtain the VSM-Diffusion, as shown in Figure 1(d). Distinct from the hierarchy of latent variables in VSM-VAE, we share the same context and memory variables across all hidden layers of a diffusion model, specifically $c_i = c_i^{(l)}$ and $m_i = m_i^{(l)}$ for $l \in \{1, ..., L\}$. Thus, under the generative formulate of MNS, using the same symbol notation in Eq. (1), the per-sample generative process of $x_{i,j}$ in the dataset $D_i$ can be formulated as

$$p_\theta(x_{i,j}) = \int \int \int p_\theta(x_{i,j} \mid \{x_{i,j}^{(l)}\}_{l=1}^{L}, c_i)p_\theta(c_i \mid m_i)p(m_i)\,d\{x_{i,j}^{(l)}\}_{l=1}^{L}\,dc_i\,dm_i,$$

where the backward diffusion process is consistent with that of conditional diffusion model [24, 25], formulated as

$$p_\theta(\boldsymbol{x}_{i,j} \mid \{\boldsymbol{x}_{i,j}^{(l)}\}_{l=1}^L, \boldsymbol{c}_i) = p_\theta(\boldsymbol{x}_{i,j} \mid \boldsymbol{x}_{i,j}^{(1)}, \boldsymbol{c}_i) \prod_{l=1}^{L-1} p_\theta(\boldsymbol{x}_{i,j}^{(l)} \mid \boldsymbol{x}_{i,j}^{(l+1)}, \boldsymbol{c}_i) p_\theta(\boldsymbol{x}_{i,j}^{(L)}). \tag{8}$$

**Training objective of VSM-Diffusion:** For VSM-Diffusion, the training objective is derived from maximizing the evidence lower bound (ELBO) of a set of training datasets $\{D_i\}_{i=1}^N$, which is consistent with Eq. (5). Similar to regular DDPMs, the first term can be decomposed into a summation of $L$ terms, each of which can be computed independently, and can be estimated as

$$\mathbb{E}_Q\left[\log p_\theta(D_i \mid \boldsymbol{c}_i, \boldsymbol{m}_i)\right] = \sum_{j=1}^{J_i} \mathbb{E}_Q\left[-\log \frac{p_\theta(\boldsymbol{x}_{i,j}, \{x_{i,j}^{(l)}\}_{l=1}^L \mid -)}{q_\phi(\{x_{i,j}^{(l)}\}_{l=1}^L \mid \boldsymbol{x}_{i,j})}\right] = \sum_{j=1}^{J_i} L_{j,0} + \sum_{t=2}^{L} L_{j,l-1} + L_{j,L},$$

where $L_L$ is a fixed constant, $L_0$ is the expectation of likelihood term and the intermediate term $L_{l-1}$ can be formulated as

$$L_{j,l-1} = \mathbb{E}_{q_\phi\left(x_{i,j}^{(l)} \mid \boldsymbol{x}_{i,j}\right)} \left[\log \frac{q_\phi(x_{i,j}^{(l-1)} \mid x_{i,j}^{(l)}, \boldsymbol{x}_{i,j})}{p_\theta(x_{i,j}^{(l-1)} \mid x_{i,j}^{(l)}, \boldsymbol{c}_i)}\right]. \tag{9}$$

## 4 Variational Structured Memory

In this section, we will introduce the storage architecture of VSM in Section 4.1, equipped with the mechanism of memory recall and update in Sections 4.2 and 4.3, respectively. The basic intuition of VSM is inspired by the memory mechanism in the human brain: 1) there are two advanced forms of memory in the human brain: semantic memory allows the storage of general conceptual information and episodic memory allows the collection of detailed episodes; 2) episodic memory can provide detailed episodes that will be converted into conceptual information and stored in semantic memory. The structured memories stored in VSM will be recalled to infer the posterior of task-specific memory, formulated as:

$$q_\phi(\boldsymbol{m}_i \mid D_i, \mathbf{M}) \tag{10}$$

which will be used in the inference process of either VSM-VAE or VSM-Diffusion.

### 4.1 Structured Memory Storage

As shown in Figure 7, there are $N$ blocks (categories) in the VSM module, denoted as $\mathbf{M} = \{\mathbf{M}_n\}_{n=1}^N$, and each memory block $\mathbf{M}_n$ consists of both semantic memory $\mathbf{M}_n^{(s)}$ and episodic memory $\mathbf{M}_n^{(e)}$.

**Semantic Memory Storage:** The semantic memory in the human brain can provide conceptual information for quickly learning concepts of new object categories by seeing only a few data samples. Following Zhen et al. [18], for each block (category) of semantic memory in VSM, we try to store the conceptual information by averaging the latent representations of data samples belonging to this category, contributing to lightweight storage and fast lookup. To model the uncertainty, we model the $n$-th block (category) of semantic memory with a Gaussian distribution, formulated as:

$$\mathbf{M}^{(s)} = \{\mathbf{M}_n^{(s)}\}_{n=1}^N, \quad \mathbf{M}_n^{(s)} = \mathcal{N}(\boldsymbol{\mu}_n, \mathrm{diag}(\boldsymbol{\sigma}_n)), \tag{11}$$

where $\boldsymbol{\mu}_n, \boldsymbol{\sigma}_n \in \mathbb{R}^d$ denotes the $d$-dimensional mean and variance vectors, respectively.

**Episodic Memory Storage:** Distinct from semantic memory allowing the storage of general conceptual information, episodic memory tends to focus on storing more detailed and vivid experiences or episodes. The design of episodic memory is inspired by previous works [26–30], and the main idea is to employ a query memory module to collect a subset of representative data samples for each block (category) in VSM. Specifically, the $n$-th block (category) of episodic memory is a set of $T$ embedding vectors of data samples:

$$\mathbf{M}^{(e)} = \{\mathbf{M}_n^{(e)}\}_{n=1}^N, \quad \mathbf{M}_n^{(e)} = \{\mathbf{M}_{n,t}^{(e)}\}_{t=1}^T, \tag{12}$$

where $T$ is the storage limit of each block (category) of episodic memory, and the dimension of $\mathbf{M}_{n,t}^{(e)} \in \mathbb{R}^d$ is the same as the latent representations of data samples.

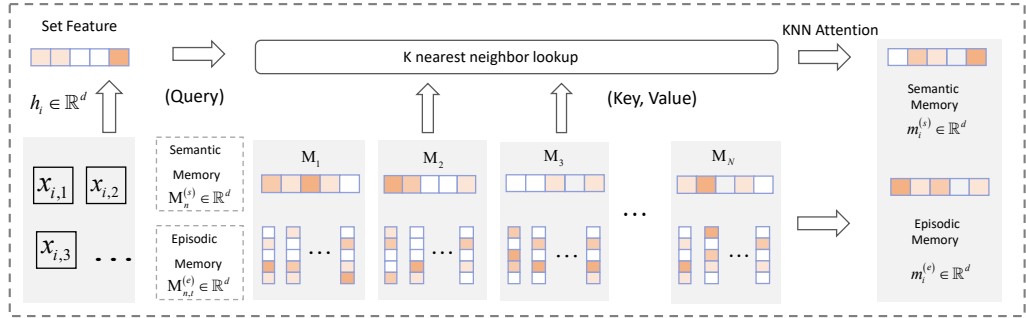

Figure 2: The workflow of memory recall mechanism in variational structured memory (VSM) module $\mathbf{M}$, which stores both semantic memory $\mathbf{M}^{(s)}$ and episodic memory $\mathbf{M}^{(e)}$.

## 4.2   Structured Memory Recall

Given a query dataset $D_i = \{x_{i,j}\}_{j=1}^{J_i}$, which can be treated as a new dataset $D_{new}$ for previous tasks, the mechanism of memory recall aims at effectively using the memory module to quickly adapt a model trained on previous tasks to a new unseen task. The workflow of structured memory recall in VSM has been depicted in Figure 7, which includes both the recalls of semantic memory $\mathbf{M}^{(s)}$ and episodic memory $\mathbf{M}^{(e)}$, resulting in two kinds of task-specific memories, denoted as $\boldsymbol{m}_i^{(s)}$ and $\boldsymbol{m}_i^{(e)}$ respectively.

**Semantic Memory Recall:** For a new few-shot generation task on the dataset $D_i$ ($D_{new}$), the memory recall in VSM will first select the top-K relevant blocks (categories) of semantic memory as the candidates, denoted as $\{\mathbf{M}_k^{(s)}\}_{k=1}^K$ , and then use an attention mechanism to aggregate these candidates into a task-specific semantic memory $\boldsymbol{m}_i^{(s)}$, where the attention weight of $n$-th can be calculated as

$$g\left(\mathbf{M}_n^{(s)}, \boldsymbol{h}_i\right) = \exp\left(\boldsymbol{\mu}_n - \boldsymbol{h}_i\right)^2 / 2\boldsymbol{\sigma}_n^2, \tag{13}$$

where $g(\cdot)$ can be treated as a similarity function, and $\boldsymbol{h}_i$ indicates the semantic representation of dataset $D_i$ and can be obtained by averaging the latent representations of data samples in $D_i$.

Then, with the top-K relevant blocks of semantic memory $\{\mathbf{M}_k^{(s)}\}_{k=1}^K$ in hand, we can further aggregate these candidates with their attention weights, formulated as:

$$\boldsymbol{m}_i^{(s)} = \sum_{k=1}^K \boldsymbol{\lambda}_{i,k}\mathbf{M}_k^{(s)}, \ \boldsymbol{\lambda}_{i,k} = \frac{g\left(\mathbf{M}_k^{(s)}, \boldsymbol{h}_i\right)}{\sum_{k=1}^K g\left(\mathbf{M}_k^{(s)}, \boldsymbol{h}_i\right)}, \ \mathbf{M}_k^{(s)} \sim \mathcal{N}(\boldsymbol{\mu}_k, \mathrm{diag}(\boldsymbol{\sigma}_k)), \tag{14}$$

where $\boldsymbol{m}_i^{(s)} \in \mathbb{R}^d$ has gathered the relevant information from the semantic memory $\mathbf{M}^{(s)}$ for the generation task on dataset $D_i$.

**Episodic Memory Recall:** After selecting the top-K relevant blocks of semantic memory $\{\mathbf{M}_k^{(s)}\}_{k=1}^K$, we can further dive into recalling episodic memory, which is more specific and detailed. In practice, each block of episodic memory contains $T$ embedding vectors of data samples $\mathbf{M}_k^{(e)} = \{\mathbf{M}_{k,t}^{(e)}\}_{t=1}^T$, and we can finally obtain $K * T$ embedding vectors of data samples. After that, following a similar process of recalling episodic memory, we can further aggregate these obtained embedding vectors with their attention weights as

$$\boldsymbol{m}_i^{(e)} = \sum_{k=1}^K \sum_{t=1}^T \boldsymbol{\lambda}_{i,k,t}\mathbf{M}_{k,t}^{(e)}, \quad \boldsymbol{\lambda}_{i,k,t} = \frac{g\left(\mathbf{M}_{k,t}^{(e)}, \boldsymbol{h}_i\right)}{\sum_{k=1}^K \sum_{t=1}^T g\left(\mathbf{M}_{k,t}^{(e)}, \boldsymbol{h}_i\right)}, \tag{15}$$

where $g(\cdot)$ and $\boldsymbol{h}_i$ have been defined in Eq. (13), and the obtained $\boldsymbol{m}_i^{(e)} \in \mathbb{R}^d$ has gathered the relevant information from the episodic memory $\mathbf{M}^{(e)}$ for the generation task on dataset $D_i$.

### 4.3 Structured Memory Update and Consolidation

**Semantic Memory Update:** Inspired by the characteristics of human memory [31–33], we develop a novel semantic memory update mechanism that considers the data information from previous semantic memory $\mathbf{M}^{(s)}$, episodic memory $\mathbf{M}^{(e)}$, and new task $D_i$. Specifically, the semantic memory module is empty at the beginning of learning, and each unseen block (category) $\mathbf{M}_n^{(s)}$ will be initialized with the mean and variance of the latent representations of data samples, which is collected from the current task $D_i$ and belongs to $n$-th category. For those seen categories, their semantic memory can be updated with the data sample from the current task $D_i$ using self-attention mechanism.

Following Zhen et al. [18], we construct a graph with respect to the memory $\mathbf{M}_n^{(s)}$ for updating, which consists of latent representations of $J_n$ data samples that belongs to $n$-th category and $T$ data samples stored in episodic memory $\mathbf{M}_n^{(e)}$, denoted as $H_n = \{\boldsymbol{h}_{n,1}, \boldsymbol{h}_{n,2}, \cdots, \boldsymbol{h}_{n,J_n}, \mathbf{M}_{n,1}^{(e)}, \cdots, \mathbf{M}_{n,T}^{(e)}\}$. The mean of the semantic memory $\mathbf{M}_n^{(s)}$ can be updated as:

$$\boldsymbol{\mu}_{\mathbf{M}_n^{(s)}}^{new} \leftarrow \alpha \boldsymbol{\mu}_{\mathbf{M}_n^{(s)}}^{old} + (1-\alpha)\,\bar{\boldsymbol{\mu}}_{\mathbf{M}_n^{(s)}}, \; \bar{\boldsymbol{\mu}}_{\mathbf{M}_n^{(s)}} = \sum_{j=1}^{J_n+T} \beta_{n,j} H_{n,j}, \; \beta_{n,j} = \frac{g\left(\mathbf{M}_n^{(s)}, H_{n,j}\right)}{\sum_{j=1}^{J_n+T} g\left(\mathbf{M}_n^{(s)}, H_{n,j}\right)}, \quad (16)$$

where $\alpha \in (0,1)$ is a hyperparameter; $\beta_{n,j}$ indicates the attention weight between $j$-th node features and the source node $\mathbf{M}_n^{(s)}$; $g(\cdot)$ is the similarity function defined in Eq. (13). This operation allows the useful information to be retained in memory while erasing less relevant or trivial information.

After that, we employ the incremental update strategy of the Gaussian distribution [34] to update the variance of semantic memory $\mathbf{M}_n^{(s)}$ as follows:

$$\boldsymbol{\sigma}_{\mathbf{M}_n^{(s)}}^{2\,new} \leftarrow \alpha \left[ \boldsymbol{\sigma}_{\mathbf{M}_n^{(s)}}^{2\,old} + (\boldsymbol{\mu}_{\mathbf{M}_n^{(s)}}^{new} - \boldsymbol{\mu}_{\mathbf{M}_n^{(s)}}^{old})^2 \right] + (1-\alpha)\left[ \boldsymbol{\sigma}_{H_n}^2 + (\boldsymbol{\mu}_{\mathbf{M}_n^{(s)}}^{new} - \bar{H}_n)^2 \right], \quad (17)$$

where $\bar{H}_n$ and $\sigma_{H_n}^2$ are the mean and variance of the set of node features $H_n$, respectively.

**Episodic Memory Update:** Inspired by the fact that it is difficult for a person to forget those things he/she often recalls (and vice versa), we design a frequency-based episodic memory update stragy. Typically, for each data sample $\mathbf{M}_{n,t}^{(e)}$ stored in episodic memory $\mathbf{M}_n^{(e)}$, we use a frequency matrix, denoted as $C_{\mathbf{M}_{n,t}^{(e)}}$, to record how many times $\mathbf{M}_{n,t}^{(e)}$ was recalled during training. Then, for updating $\mathbf{M}_n^{(e)} = \{\mathbf{M}_{n,t}^{(e)}\}_{t=1}^T$, if the current episodic memory block has not been filled yet, we will directly append the latent representation of a new data sample to the $\mathbf{M}_n^{(e)}$ when a new task arrives; Otherwise, we will update $\mathbf{M}_n^{(e)}$ by replacing the least used data sample in the block if its capability exceeds the limit $T$. We note that the frequency matrix $C_{\mathbf{M}_{n,t}^{(e)}}$ will be reset as the zero matrix after each update of episodic memory. The detailed memory updating process algorithm can be found in Appendix A.

## 5 Experiments

### 5.1 Experimental setup

**Datasets:** The experiments are conducted on five widely used benchmark datasets of various sizes, including binarized Omniglot [2], MNIST [35], DOUBLE-MNIST [36], CelebA [37] and FS-CIFAR100 [38]. The split of training/testing set (also known as background-evaluation split) follows Lake et al. [2], in which scenario all the samples at test time are from new classes. We resize all the images of binary datasets to 28x28, FS-Cifar100 to 32x32, and CelebA to 64x64.

**Baselines:** 1) For VAE-based models, we utilize NS-related models, specifically focusing on two NS variants: Convolutional Neural Statistician (CNS), where the latent space is shaped with convolutions at a given resolution; Set-Context-Hierarchical-Aggregation-Variational-Autoencoder (SCHA-VAE) [24], which employs an additional hierarchy over the context latent variable $c$. The CNS and SCHA-VAE can be naturally extended under the framework of VSM-VAE, leading to VSM-CNS and VSM-SCHA, respectively. Meanwhile, we create several variants by fusing either semantic memory or episodic memory with CNS or SCHA-VAE, as appropriate, to further demonstrate the efficacy of the proposed variational structured memory; 2) For Diffusion-based models, we consider both

Table 1: Few-shot generative evaluation of VAE-based models on various datasets with the set size 5, including Omniglot, Double-Mnist, FS-Cifar100, and Mnist (trained on Omniglot and tested on Mnist.). For all datasets, we set the episodic memory size of all methods to 5 and the semantic memory size to be the same as the number of categories appearing in training data.

| | Omniglot-ns | | Double-mnistl | | Mnist | | FS-Cifar100 | |
| | NELBO | NLL | NELBO | NLL | NELBO | NLL | NELBO(bpd) | NLL(bpd) |
|---|---|---|---|---|---|---|---|---|
| VAE | 101.5 | 95.9 | 74.3 | 69.2 | 125.1 | 105.1 | - | - |
| NS | 96.6 | 92.4 | 67.3 | 60.2 | 118.7 | 101.2 | - | - |
| CNS | 92.9 | 89.7 | 66.5 | 54.3 | 114.1 | 90.4 | 9.979 | 9.803 |
| CNS + Semantic | 84.4 | 72.4 | 56.7 | 50.2 | 107.3 | 88.5 | 9.952 | 9.745 |
| CNS + Episodic | 83.9 | 71.8 | 54.3 | 51.1 | 97.8 | 80.3 | 9.946 | 9.741 |
| VSM-CNS(our) | 81.1 | 68.2 | 47.8 | 40.6 | 92.9 | 77.0 | **9.872** | **9.711** |
| SCHA-VAE | 89.4 | 85.8 | 68.1 | 53.3 | 114.7 | 88.5 | 10.364 | 10.338 |
| SCHA + Semantic | 81.5 | 66.8 | 53.7 | 48.9 | 98.5 | 77.7 | 10.317 | 10.215 |
| SCHA + Episodic | 82.5 | 63.4 | 53.0 | 47.2 | 95.3 | 79.1 | 10.292 | 10.176 |
| VSM-SCHA(our) | **74.0** | **56.8** | **42.4** | **36.5** | **89.9** | **68.2** | 10.219 | 10.041 |

Table 2: Few-shot generative evaluation of diffusion-based models on various datasets with the set size 5, including Fs-Cifar100 and Celeba. (FID: Frechet score; sFID: spatial FID; P: precision; R: recall.)

| | FS-Cifar100 | | | | Celeba | | | |
| | FID ↓ | sFID ↓ | P ↑ | R ↑ | FID ↓ | sFID ↓ | P ↑ | R ↑ |
|---|---|---|---|---|---|---|---|---|
| DDPM | 62.84 | 28.91 | 0.58 | 0.40 | 14.02 | 27.46 | 0.71 | 0.39 |
| sDDPM | 45.50 | 29.87 | 0.54 | 0.46 | 12.87 | 26.44 | 0.72 | 0.37 |
| vDDPM | 62.58 | 27.50 | 0.58 | 0.41 | 13.01 | 26.24 | **0.73** | 0.39 |
| vFSDM | 63.73 | 28.85 | 0.58 | 0.38 | 13.52 | 27.75 | 0.71 | 0.38 |
| VSM-vFSDM(our) | **44.11** | **23.11** | **0.59** | **0.46** | **12.27** | **25.79** | 0.72 | **0.40** |
| FSDM | 35.07 | 20.95 | 0.62 | 0.53 | 10.11 | 24.58 | 0.73 | 0.39 |
| VSM-FSDM(our) | **33.21** | **18.32** | **0.68** | **0.61** | **9.25** | **23.21** | **0.77** | **0.43** |

unconditional and conditional diffusion models as baselines, where we use the DDPM [39] as the unconditional baseline and other three conditional diffusion models: VDDPM, where the context $c$ is a latent variable and the dataset $D$ is generated conditioned on $c$; VFSDM [24], another variational diffusion model where employees a different way to extract and aggregate set-information using ViT [40]: stack all the samples on the channel dimension; sDDPM adapting ideas in Sinha et al. [41] without contrastive learning, where a ViT encoder splits the image in patches and processes them jointly.

**Model Settings:** The details about model settings can be found in Appendix B.

## 5.2 Quantitative experiment

**VAE-based models**: To evaluate the generative properties of VAE-based models, we use the evidence lower bound (ELBO) to approximate the log marginal likelihood with 1000 importance samples (NLL). Our qualitative experiments are performed on Omniglot, DOUBLE-MNIST, MNIST, and FS-Cifar100. The outcomes of the experiment are displayed in Table 5.2. It is clear that VSM-CNS/VSM-SCHA can achieve significant improvements over its baseline models, proving that VSM could be a promising mechanism for few-shot generation. Additionally, by incorporating either semantic or episodic memory into CNS/SCHA-VAE, the resulting models can also achieve better performance, proving both memory types can improve the model performance of few-shot generation.

**Diffusion-based models:** As exhibited in Table 5, we employ the metrics below to assess the generative capacity of diffusion-based models, including: FID [42] for sample quality, sFID [43] to capture spatial relationships, and Precision and Recall [44] for measuring variety and mode coverage. From the results, VSM-Diffusion beats the other unconditional and conditional baselines on both two datasets, demonstrating the effectiveness of VSM on few-shot generation.

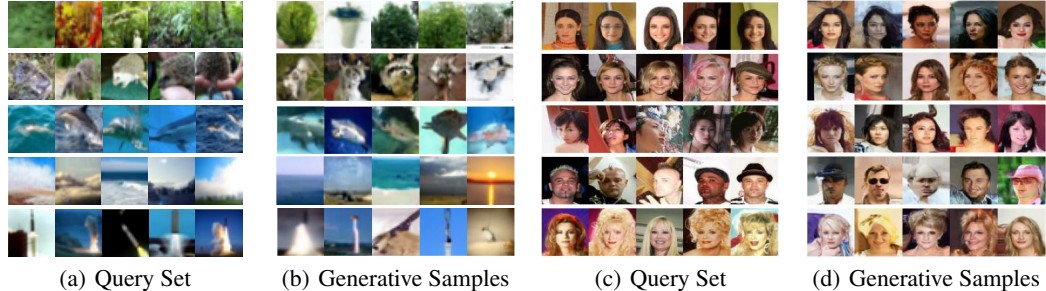

|      (a) Query Set      |    (b) Generative Samples    |      (c) Query Set      |    (d) Generative Samples    |

Figure 3: Visualization of few-shot generative samples using a VSM-Diffusion with the set size 5. The query set and generative samples on the left side are from FS-Cifar100, while those on the right side are from Celeba.

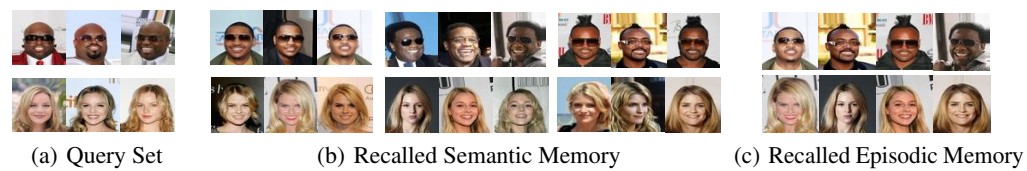

|   (a) Query Set   |   (b) Recalled Semantic Memory   |   (c) Recalled Episodic Memory   |

Figure 4: (a) Query Set; (b) Recalled blocks (categories) of semantic memory, each of which is represented by the images of corresponding category; (c) Recalled samples of episodic memory.

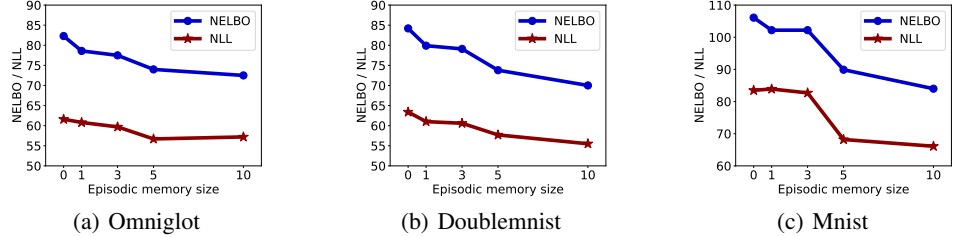

|   (a) Omniglot   |   (b) Doublemnist   |   (c) Mnist   |

Figure 5: Few-shot generative evaluation of VAE-based models on various datasets with different episodic memory sizes.

## 5.3 Qualitative experiment

**Few-shot image generation:** The generation results of VSM-Diffusion are exhibited in Figure 3. From the results, we can find that our memory-augmented models can generate class-specific samples through successfully extracting content information and realistic classes that are both complicated and varied. Limited by pages, additional few-shot generative experimental results on Omniglot, CelebA, and FS-Cifar100 can be found in Appendix C.2.

Table 3: Ablation study on the update strategy of variational structured memory (VSM).

|  | Omniglot-ns | | Double-Mnist | | Mnist | |
|---|---|---|---|---|---|---|
|  | NELBO | NLL | NELBO | NLL | NELBO | NLL |
| W/o Episodic memory | 75.4 | 58.2 | 45.3 | 40.1 | 91.1 | 69.5 |
| W/o Frequency-based | 74.8 | 58.2 | 42.9 | 37.7 | 91.7 | 70.6 |
| **Combined strategy** | **74.0** | **56.7** | **42.4** | **36.5** | **89.9** | **68.2** |

**Memory visualization:** To further verify the effectiveness of the proposed VSM, we conduct experiments of memory visualization here. For instance, given a new set, as shown in Figure 4(a), the images of recalled semantic memory corresponding to the class are shown in Figure 4(b), and the images of recalled episode memory are shown in Figure 4(c). The visualization results demonstrate that VSM can recall related image classes and samples to the query set in an effective way.

### 5.4 Ablation study

**Effect of memory size:** To investigate the affect of memory size in episodic memory, we use the VSM-SCHA model as a typical example. We train a model on Omniglot data, and evaluate it with the Omniglot, DOUBLE-MNIST, and MNIST datasets for unknown classes. The size of the episodic memory is raised from 0 to 10. The results in Figure 5 show that the model performance improves with the increase of episodic memory size, further demonstrating the importance of episodic memory in few-shot generation tasks. We note that the developed variational structured memory will reduce to variational semantic memory [18] when the episodic memory size is adjusted to 0.

**Effect of memory update process:** We conduct two variants of VSM-SCHA to investigate the benefits of the proposed structured memory update strategy. For the first variant, its semantic memory update strategy does not consider episodic memory, denoted as *W/o Episodic memory*; For the second variant, its episodic memory update strategy does not employ a frequency-based strategy but uses a first-in, first-out queue-based memory update method, denoted as *W/o Frequency-based*. The experiment results in Table 3 can verify the effectiveness of the proposed memory update strategy.

## 6  Conclusion

In this study, we develop a novel variational structured memory module that can concurrently retain generic semantic structure about a category and specific information about a sample, the former being achieved by semantic memory and the latter being caught by episodic memory. Meanwhile, we introduce a bionic memory updating strategy for the conversion between episodic and semantic memories stored in VSM, where episodic memory can provide context information that will later be transferred and stored in semantic memory. Through incorporating VSM with few-shot latent variable models, we develop a novel hierarchical memory-augmented neural statistician (HMNS), whose generative process can be specified with a VAE or diffusion model, leading to VSM-VAE and VSM-Diffusion. Extensive experiments demonstrate that our developed VSM-VAE and VSM-Diffusion can achieve promising performance on few-shot generation tasks.

## Acknowledgments

This project was supported in part by the National Natural Science Foundation of China under Grant U21B2006; in part by Shaanxi Youth Innovation Team Project; in part by the Fundamental Research Funds for the Central Universities QTZX23037 and QTZX22160; in part by the 111 Project under Grant B18039; in part by the Fundamental Research Funds for the Central Universities; in part by the Innovation Fund of Xidian University under Grant YJSJ23016; Additionally, this project was supported in part by the National Research Foundation, Singapore under its Industry Alignment Fund – Pre-positioning (IAF-PP) Funding Initiative.

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

# A  Memory Recall and Update Algorithm

The procedure of memory recall and update algorithm is summarized in Algorithm. 1.

---

**Algorithm 1** Semantic Memory recall and update algorithm

---

**Input**: Given task $D_i$; Semantic memory $\mathbf{M}^{(s)}$, and episodic memory $\mathbf{M}^{(e)}$;
Recall task-specific semantic memory according to Eq. 14 and episodic memory according to Eq. 15 ;
**if** The semantic memory for the given task is null: **then**
    Computing the mean and variance for the semantic memory with given task $D_i$;
**else**
    Update the semantic memory according to Eq. 16 and Eq. 17;
**end if**
**if** The episodic memory for the given task is null: **then**
    Append episodic memory with given task $D_i$;
**else**
    Update the episodic memory according to frequency matrix $C$;
**end if**
Infer variational posteriors for the latent variables of different layers $\{\boldsymbol{\theta}^{(l)}\}_{1=1}^{L}$;

---

**Model Design:** In most memory-based models, the images $\boldsymbol{D}_i = \{\boldsymbol{x}_{ij}\}_{j=1}^{J_i}$ are fed into an encoder to produce image features $\{h_{ij}\}_{j=1}^{J_i} \in \mathbb{R}^{D \times 1}$, which are then represented as vectors. These vector representations are utilized for both recall and update operations in the model. Nonetheless, due to the high dimensionality of images, relying solely on vectors can lead to a loss of valuable information.By drawing inspiration from the approach proposed in the Vision Transformer, we can overcome these challenges and obtain more accurate memory information. We begin by obtaining image features, denoted by $\{h_{ij}\}_{j=1}^{J_i} \in \mathbb{R}^{D \times H \times W}$, then calculate the average of the set dimension resulting in $h_i = \frac{1}{J_i} \sum_{j=1}^{J_i} h_{ij}$ Afterwards, we reshape the size of the image features to $\{h_i\} \in \mathbb{R}^{N \times D}$where $D$ denotes the patch size.When updating $M_n^{(s)} \in \mathbb{R}^{N \times D}$, we employ the resized feature $h_i^m \in \mathbb{R}^{(J_i \times N) \times D}$ for graph attention. As for $M_n^{(e)} \in \mathbb{R}^{L \times N \times D}$, where $L$ denotes the total number of episodic memory, we directly replace the least recently used patch in $M_n^{(e)}$ with $h_i^m$.By using per-patch recall and update, we obtain information from different semantic levels. This makes the Semantic Memory more suitable for task generalization, while the Episodic Memory is more capable of capturing sample-specific details.

**VSM-SCHA** In SCHA-VAE and VSM-SCHA, we use a shared routing path between c and z: $q_\phi(c_i^{(l)}|D_i, c_i^{(l+1)}, z_i^{(l+1)}, m_i^{(l)})$ where $z_{l+1}$ and $c_{l+1}$ are samples from layer $l$. We additionally remove any form of normalization (batch/layer norm) in the hierarchical formulation

**Memory size** Across all datasets, we set the size of each semantic memory block (category) in the VSM-based model to 1, and each episodic memory block to 5. For models that exclusively depend on episodic memory (CNS + Episodic, SCHA + Episodic, and Episodic-diffusion), we establish a memory size of 10000, with the episodic memory functioning as a first-in, first-out queue.

# B  Details of Model Design

# C  Additional Experiment Results

## C.1  Quantitative Comparison of Generative Samples

In this section, we apply VSM to FSDM, a conditional diffusion model where c is not a variational variable. To align with FSDM, we only use the following update rule for the semantic memory:

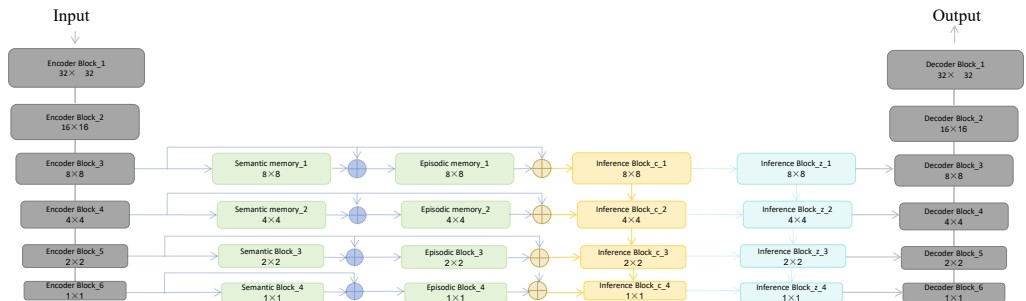

Figure 6: The process of memory recall mechanism in VSM-SCHA.

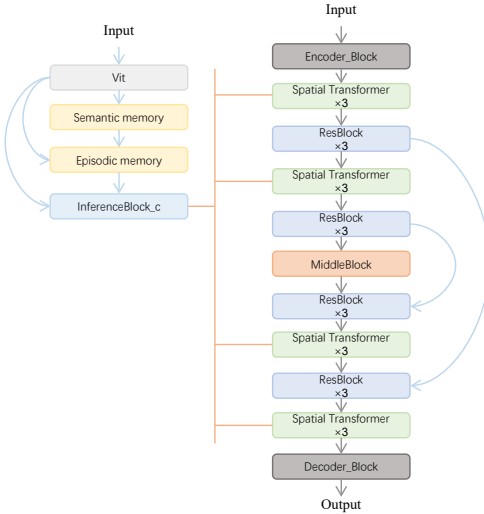

Figure 7: The process of memory recall mechanism in VSM-FSDM.

$$\boldsymbol{\mu}_{\mathbf{M}_n^{(s)}}^{new} \leftarrow \alpha \boldsymbol{\mu}_{\mathbf{M}_n^{(s)}}^{old} + (1-\alpha)\, \bar{\boldsymbol{\mu}}_{\mathbf{M}_n^{(s)}}, \; \bar{\boldsymbol{\mu}}_{\mathbf{M}_n^{(s)}} = \sum_{j=1}^{J_n+T} \beta_{n,j} H_{n,j}, \; \beta_{n,j} = \frac{g\left(\mathbf{M}_n^{(s)}, H_{n,j}\right)}{\sum_{j=1}^{J_n+T} g\left(\mathbf{M}_n^{(s)}, H_{n,j}\right)} \quad (18)$$

Table 4 presents the results of VSM-FSDM on Fs-Cifar100 and celeba datasets. It can be observed that VSM performs well even without using variational latent variables, demonstrating its strong performance and scalability.

## C.2 Additional Visualization of Generative Samples

More visualization results of generative samples can he found in Figures 8, 9, 10 and 11.

Table 4: Few-shot generative evaluation of diffusion-based models on various datasets with the set size 5, including Fs-Cifar100 and Celeba. (FID: Frechet score; sFID: spatial FID; P: precision; R: recall.)

|  | FS-Cifar100 | | | | Celeba | | | |
|---|---|---|---|---|---|---|---|---|
|  | FID ↓ | sFID ↓ | P ↑ | R ↑ | FID ↓ | sFID ↓ | P ↑ | R ↑ |
| DDPM | 62.84 | 28.91 | 0.58 | 0.40 | 14.02 | 27.46 | 0.71 | 0.39 |
| FSDM | 35.07 | 20.95 | 0.62 | 0.53 | 10.11 | 24.58 | 0.73 | 0.39 |
| VSM-FSDM | **33.21** | **18.32** | **0.68** | **0.61** | **9.25** | **23.21** | **0.77** | **0.43** |

Table 5: Ablation study on diffusion-based models

| | FS-Cifar100 | | | | Celeba | | | |
|---|---|---|---|---|---|---|---|---|
| | FID ↓ | sFID ↓ | P ↑ | R ↑ | FID ↓ | sFID ↓ | P ↑ | R ↑ |
| vFSDM | 63.73 | 28.85 | 0.58 | 0.38 | 13.52 | 27.75 | 0.71 | 0.38 |
| vFSDM + semantic | 60.21 | 27.63 | 0.58 | 0.42 | 12.98 | 26.01 | 0.73 | 0.39 |
| vFSDM + episodic | 52.13 | 25.25 | 0.59 | 0.44 | 12.36 | 25.97 | 0.72 | 0.41 |
| VSM-diffusion | **44.11** | **23.11** | **0.59** | **0.46** | **12.27** | **25.79** | 0.72 | **0.40** |

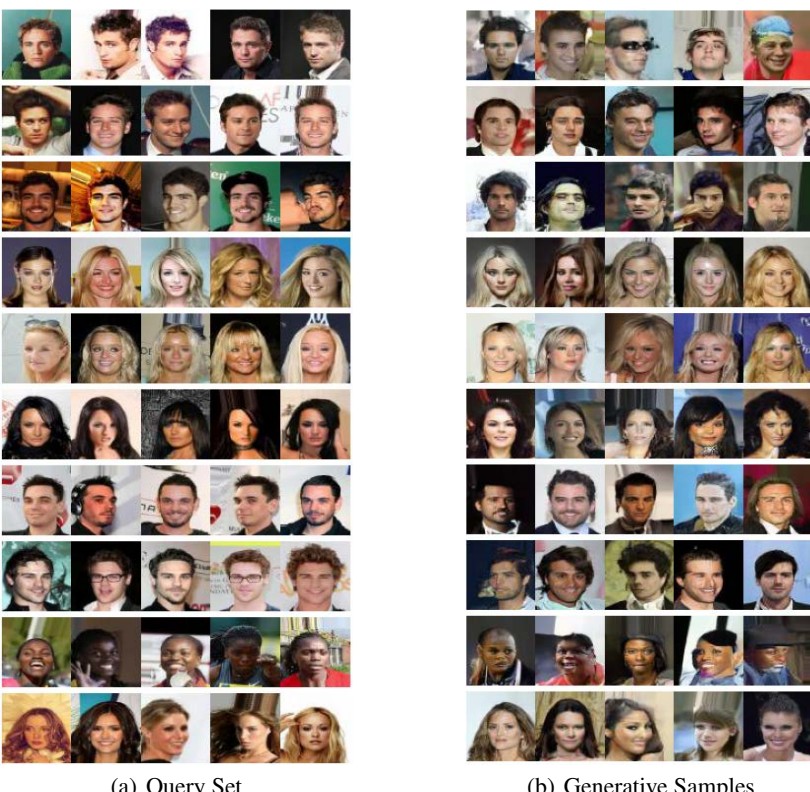

(a) Query Set    (b) Generative Samples

Figure 8: Visualization of few-shot generative samples using a VSM-Diffusion with the set size 5 on Celeba dataset. (Conditioning set and samples from in-distribution classes).

# D Border Impact

The impact of our research can be highlighted in two key aspects:

**For Few-shot Learning Task:** Our structured memory has demonstrated a high level of effectiveness in storing and managing task-related information, enabling efficient transfer of prior knowledge from previous tasks to new ones. By combining semantic and episodic information storage, our model enhances its ability to capture both task-level information and fine-grained image-level details. Diverging from traditional few-shot models, our approach leverages the stored and recalled memory to extract more valuable insights while minimizing the influence of irrelevant information on new tasks.

**For Deep Generative Models:** Our modularized memory offers a seamless integration into a wide range of generative models, such as diffusion models and VAEs, while also easily merging with latent variables. This flexibility allows for efficient utilization of the memory component within the generative framework. In addition, our structured memory can be divided into multiple semantic levels, providing the ability to learn at various refined semantic layers. This hierarchical organization of memory facilitates capturing complex dependencies and relationships across different levels of

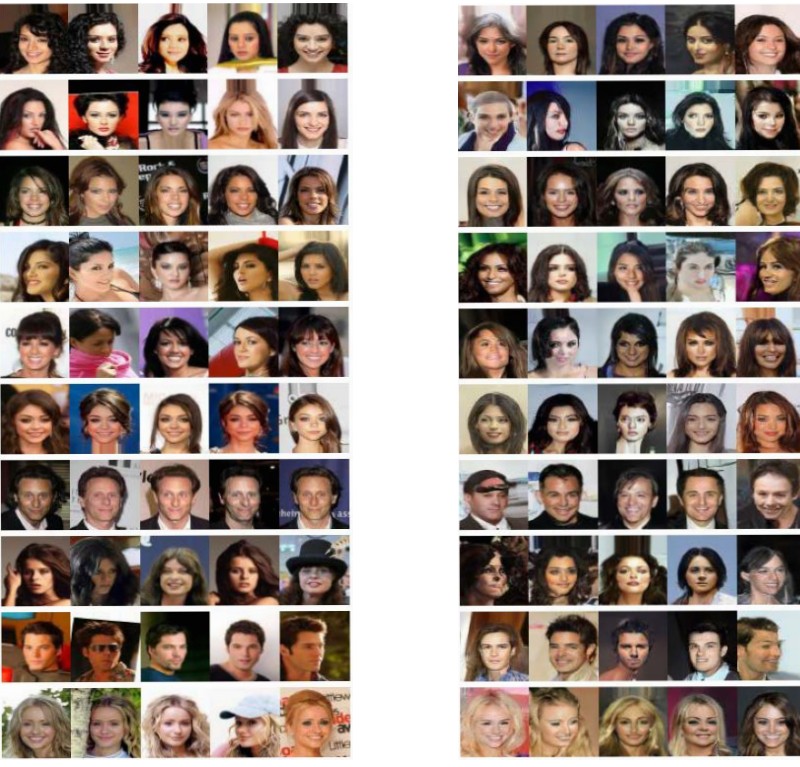

(a) Query Set              (b) Generative Samples

Figure 9: Visualization of few-shot generative samples using a VSM-Diffusion with the set size 5 on Celeba dataset. (Conditioning set and samples from out-distribution classes)

abstraction. By incorporating multiple semantic layers, generative models can potentially achieve enhanced performance and generate more meaningful and coherent outcomes.

## E  Limitation

The proposed model variant, VSM-Diffusion, is based on the denoise diffusion model, which requires multiple steps to generate samples. The framework can be extended to other diffusion models, such as implicit denoise diffusion models, to accelerate the generative process and improve its efficiency.

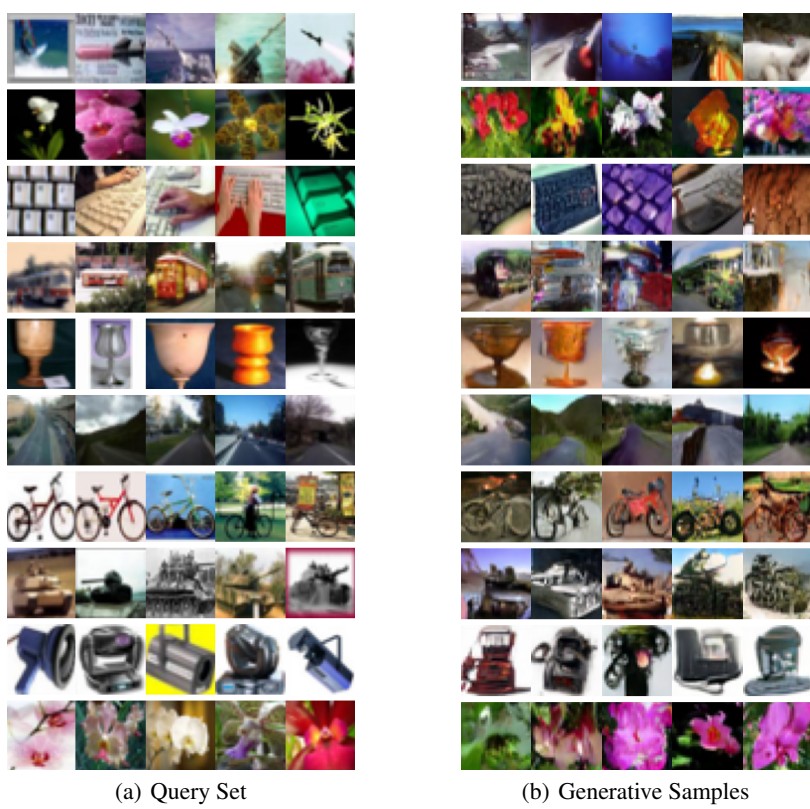

(a) Query Set         (b) Generative Samples

Figure 10: Visualization of few-shot generative samples using a VSM-Diffusion with the set size 5 on FS-Cifar100 dataset. (Conditioning set and samples from in-distribution classes).

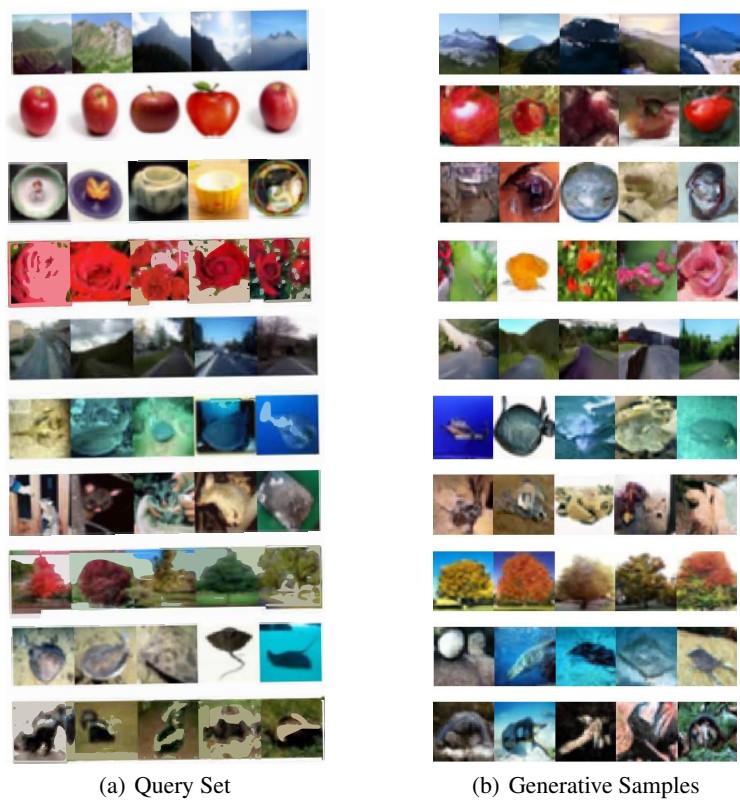

(a) Query Set          (b) Generative Samples

Figure 11: Visualization of few-shot generative samples using a VSM-Diffusion with the set size 5 on FS-Cifar100 dataset. (Conditioning set and samples from out-distribution classes).

