# OpenReview forum: "Few-shot Generation via Recalling  Brain-Inspired Episodic-Semantic Memory"
_NeurIPS.cc/2023/Conference — NeurIPS 2023 poster_

### Official Review · Reviewer_aZnV · 2023-06-30

**Soundness:** 3 good
**Presentation:** 1 poor
**Contribution:** 3 good
**Rating:** 7
**Confidence:** 3

**Summary:**

In this article the authors propose to augment state of the art few-shot generative models with a memory module inspired by the brain. The memory module combines episodic and semantic memory. The authors demonstrate through extensive experimentations the benefits of their module on few-shot generation: when taken either separately or combined, each type of memory improves the generation performance.

**Strengths:**

The claims of the article are well supported by extensive experimentations. Experimentations involves various and diversified datasets, and the baselines are well chosen. The memory module is well described with the Fig2.

**Weaknesses:**

The writing of the paper makes it unpleasant to read (despite interesting results), and should be improved: some sentences have just not meaning ! I suggest the authors to take time for an ‘in detail’ re-reading to improve the writing (I have just reported few typos, but there are plenty of them !! ).

As a general comment, the presented article seems to be scientifically correct, but I have quickly lost track of the message because the reading flow was constantly interrupted by a poor writing ! Really sad !

**Questions:**

* In the experimental section (section 5), the authors are comparing the CNS (and the SCHA-VAE) with their model with different type of memory. This comparison is very interesting, but have you controlled for the number of parameters ? On my understanding, there are more parameters in the VSM-CNS (and in the VSM-SCHA) than in the CNS (and SCHA) which makes the comparison not as informative as it should be. Would it be possible to find a way to equate the number of parameters in both type of networks ?

* Concerning the writing : some sentence are just not properly formed. Sometimes, a noun is used instead of a verbs, which make the reading really hard. For exemple : "which can simultaneously store both episodic and semantic memories to assistant existing generative models efficiently recall memories during generation. » (Line 8-9). In the core text of the article these kind of examples are numerous ! Tenses are also misused, and typos are all over the article !! Please re-read.

Here are few exemples :
Line 24-25 : A word is missing ?? \
Line 113 : Eq 9 —> Eq 5 \
Line 135 : TheMechanism —> The Mechanism \
Line 190 : initialed —> initialized \
...

**Limitations:**

The authors have correctly addressed the limitation part

---

> ### Author Rebuttal · Authors · 2023-08-10
>
> Thanks for your constructive comments and suggestions, which will be helpful for us to further improve the quality of our paper. Following your comments, we have carefully fixed these mentioned typos and also rewritten several parts of our paper for improving the experience of reading.
>
> Q1: Concern on model's parameters.
>
> A1: Yes, it is correct that introducing the proposed variational structured memory module will bring additional model parameters, leading to the amount of parameters of VSM-CNS/VSM-SCHA will be slightly larger than those of CNS/SCHA, respectively. However, we need to highlight that the amount of parameters brought by introducing the proposed variational structure memory is limited, compared to the amount of parameters of the original baselines, which has been listed in the following table.
>
> | Models | Parameters (M)| Rate |
> | ----- | -------|  ------- |
> | CNS | 7.34|  0%|
> | CNS + semantic| 7.39|  0.68% |
> | CNS + Episodic | 7.38|  0.54%|
> | VSM-CNS | 7.44| 1.36% |
>
> | Models | Parameters (M)| Rate |
> | ----- | -------|  ------- |
> | SCHA | 5.37|  0% |
> | SCHA + semantic| 5.60|  +4.28% |
> | SCHA + Episodic | 5.55|  +3.35% |
> | VSM-SCHA| 5.77|  +7.44% |
>
> | Models | Parameters (M)| Rate |
> | ----- | -------|  ------- |
> | sDDPM| 34.0|  0% |
> | vDDPM| 35.5|  4.41%|
> | vFSDM | 34.8|  2.35%|
> | VSM-Diffusion| 35.6|  4.70% |
>
> Q2: Concern on writing.
> A2: Thanks for your valuable suggestions. Following your comments, we have carefully revised our paper.
>
> For Line 8-9:
> We have modified Line 8-9 as ``Inspired by the memory mechanism of human brain, in this work, we carefully design a variational structured memory module (VSM), which can simultaneously store both episodic and semantic memories to assist existing generative models efficiently recall these memories during sample generation``
>
> For Line 24-25 : A word is missing ??
> We have modified Line 24-25 as ``While there have been promising processes for few-shot adaptation on classification tasks, less work has been done on few-shot generation [7–10], which is mainly due to the challenging nature of learning the generative process with only a few samples in an unsupervised manner [11, 12].``
>
> For Eq.9->Eq.5:
> Thanks for the notification, we have deleted Eq.9 in the manuscript.
>
> For typos:
> We have fixed these listed typos and carefully revised our paper.
>
> We appreciate your thoughtful comments on our paper once more, and we look forward to having more conversations with you.

---

> > ### Comment · Area_Chair_ZH2c · 2023-08-18
> > **Engage in the discussion with the authors**
> >
> > Dear Reviewer,
> >
> > The author has provided responses to your questions and concerns. Could you please read their responses and ask any follow-up questions, if any?
> >
> > Thank you!

---

> > ### Comment · Reviewer_aZnV · 2023-08-21
> > **Response to authors**
> >
> > Sorry for the late feedback. Thanks to the authors for the detail answer. In general, The authors have addressed my main concerns, I increase my rating to 7 (from 6 to 7).

---

### Official Review · Reviewer_jSnX · 2023-07-05

**Soundness:** 3 good
**Presentation:** 3 good
**Contribution:** 3 good
**Rating:** 7
**Confidence:** 4

**Summary:**

This paper introduces a novel approach to few-shot generation inspired by the human memory mechanism. The authors propose a Variational Structured Memory module (VSM) to store and recall episodic and semantic memories. They also introduce a bionic memory updating strategy to model the conversion between these memory types. The effectiveness of this approach is demonstrated through its integration with various existing generative models and evaluation of few-shot generation tasks.

**Strengths:**


**Originality**:
The paper presents a highly original approach by mimicking the human memory mechanism in the context of few-shot generation. The introduction of the Variational Structured Memory module (VSM) and the bionic memory updating strategy represents a creative combination of cognitive science principles with AI models.

**Quality**:
The paper is of high quality, reflected in the meticulous design of the VSM and its integration into existing generative models. The execution of the bionic memory updating strategy and its evaluation through few-shot generation tasks further attest to the paper's quality.

**Clarity**:
The paper is well-written. The authors have clearly explained their inspiration, the design of the VSM, and the results of their evaluation, making the paper accessible to readers of varying familiarity with the subject matter.

**Significance**:
The significance of this paper is considerable. By demonstrating a novel approach to the few-shot generation and opening up new directions for integrating memory mechanisms into generative models, this work contributes valuable insights to both cognitive science and artificial intelligence fields.






**Weaknesses:**

**Computation Efficiency**:
Although the integration of episodic and semantic memory in achieving few-shot generation is intriguing, the use of memory modules inherently increases the computational overhead in terms of extra storage space and prediction time. It would be valuable if the paper could address and analyze these aspects of computation efficiency.

**Acronym Overlap**:
The use of the acronym VSM for Variational Structured Memory might confuse, as it has been previously used in the same context in a paper by Zhen et al. [18]. It would be beneficial to consider using a different acronym to avoid any potential confusion for readers.

**Missing Reference and Cross-Domain adaptation**:
Indeed, the paper has overlooked the reference to the Hierarchical Variational Memory model (HVM) [1].  I would be happy if the authors could discuss whether their approach could be applied to utilize memories from different neural network layers for achieving cross-domain few-shot generation. This discussion could provide additional depth and broaden the applicability of your method.

[1] Du et al. "Hierarchical variational memory for few-shot learning across domains." ICLR 22.

**Questions:**

(1) Can the authors elaborate on how the parameter $\alpha$ impacts the model's performance?

(2) Is it feasible to apply your method to discriminative models, such as for few-shot classification tasks? Expanding the discussion to these applications would provide a broader view of your method's potential.

(3) Could the authors provide more detailed information regarding the additional model parameters introduced by your method compared to the baseline? Additionally, how does the inference time compare to the baseline? This information could provide more clarity regarding the computational efficiency of your method.

**Limitations:**

Yes.

---

> ### Author Rebuttal · Authors · 2023-08-10
>
> We appreciate your constructive comments and suggestions, which are helpful for us to improve the quality of our paper further. The concerns have been addressed below.
>
> W1: Concern about Computation Efficiency
>
> A1: Thanks, we will answer this question in our response to Q3.
>
> W2: Acronym Overlap:
>
> A2: Thanks for your valuable suggestion. We will update the acronym of our developed method from VSM to VSMM (Variational Structured Memory Module) in the revision.
>
> W3: Missing Reference and Cross-Domain adaptation
>
> A3: Thanks for bringing this excellent work into our view, and we will cite it and discuss its relevance to our work in the revision.For cross-domain adaptation, we believe that our memory-augmented method can be naturally extended for few-shot cross-domain generation task, and the most straightforward solution could be simply replacing the generative model for few-shot generation with another one for few-shot cross-domain generation as the backbone. We are willing to explore this promising direction in the furture.
>
> Q1: Can the authors elaborate on how the parameter  $\alpha$  impacts the model's performance?
>
> A1: Thanks for your suggestion, as exhibited in the following table, we have conducted the experiment to elaborate on the impact of the hyper-parameter $\alpha$, and the experimental results show that the setting of parameter $\alpha$ will not heavily influence the performance of our method.
>
> | Model    | $\alpha$ |     Onmignot |      | Double-mnist |      | Mnist |      |
> | -------- | -------- | -------- | ---- | ------------ | ---- | ----- | ---- |
> |          |          | ELBO | NLL  | ELBO | NLL  | ELBO  | NLL  |
> | VSM-CNS  | 0.1      | 83.1     | 69.5 | 49.3         | 42.7 | 93.2  | 77.6 |
> | VSM-CNS  | 0.5      | 83.2     | 69.9 | 48.1         | 42.3 | 93.5  | 77.8 |
> | VSM-CNS  | 0.7      | 81.1     | 68.2 | 47.8         | 40.6 | 92.9  | 77.0 |
> | VSM-CNS  | 0.9      | 84.1     | 70.8 | 48.7         | 42.6 | 93.9  | 78.5 |
> | VSM-SCHA | 0.1      | 74.6     | 57.0 | 42.4         | 35.6 | 89.7  | 68.1 |
> | VSM-SCHA | 0.5      | 73.4     | 56.2 | 42.2         | 35.3 | 88.4  | 67.1 |
> | VSM-SCHA | 0.7      | 74.0     | 56.8 | 42.4         | 36.5 | 89.9  | 68.2 |
> | VSM-SCHA | 0.9      | 74.8     | 57.5 | 43.8         | 37.9 | 90.1  | 69.7 |
>
> Q2: Is it feasible to apply your method to discriminative models, such as for few-shot classification tasks? Expanding the discussion to these applications would provide a broader view of your method's potential.
>
> A2: Thanks for your insightful suggestion! Actually, as your mentioned, the proposed variational structured memory module can be naturally extended from few-shot generation to other few-show discrimination tasks by simply replacing the generative models with other discriminative ones. And we believe that our method can also improve the performance of few-shot classification tasks, and will include this part of discussion about the method's potentials in our revision. Thanks again for your reminder.
>
>
> Q3: Could the authors provide more detailed information regarding the additional model parameters introduced by your method compared to the baseline? Additionally, how does the inference time compare to the baseline? This information could provide more clarity regarding the computational efficiency of your method.
>
> A3: Thanks for your suggestion, we have listed the amount of additional model parameters brought by introducing our method in the following table.
>
> | Models | Parameters (M)| Rate |
> | ----- | -------|  ------- |
> | CNS | 7.34|  0%|
> | CNS + semantic| 7.39|  0.68% |
> | CNS + Episodic | 7.38|  0.54%|
> | VSM-CNS | 7.44| 1.36% |
>
> | Models | Parameters (M)| Rate |
> | ----- | -------|  ------- |
> | SCHA | 5.37|  0% |
> | SCHA + semantic| 5.60|  +4.28% |
> | SCHA + Episodic | 5.55|  +3.35% |
> | VSM-SCHA| 5.77|  +7.44% |
>
> | Models | Parameters (M)| Rate |
> | ----- | -------|  ------- |
> | sDDPM| 34.0|  0% |
> | vDDPM| 35.5|  4.41%|
> | vFSDM | 34.8|  2.35%|
> | VSM-Diffusion| 35.6|  4.70% |
>
> We have also provided the additional inference time caused by introducing our method in the following table.
>
> | Models | Inference time(s)/(Batch_size=32)| Rate |
> | ----- | -------|  ------- |
> |CNS| | 2.77|
> |CNS + Semantic | 2.81 | +0.14 |
> |CNS + Episodic | 3.37 | +0.60|
> |VSM-CNS | 3.0924| +0.32 |
>
> | Models | Inference time(s)/(Batch_size=32)| Rate |
> | ----- | -------|  ------- |
> |SCHA-VAE | 4.17| 0 |
> |SCHA + Semantic | 4.32 | +0.15 |
> |SCHA + Episodic | 6.47 | +2.30 |
> |VSM-SCHA | 4.50| +0.33  |
>
> | Models | Inference time(s)/(Batch_size=32)| Rate |
> | ----- | -------|  ------- |
> |sDDPM | 98.52 | 0|
> |vDDPM |  92.23 | -5.67|
> |vFSDM | 103.98 | +5.46|
> |VSM-Diffusion | 107.59 | +9.07|
>
> We appreciate your insightful suggestions on how to improve our paper once more, and we look forward to having more conversations with you about this.

---

> > ### Comment · Reviewer_jSnX · 2023-08-16
> >
> > I thank the authors for their careful responses, which addressed most of my concerns. I hope the authors include the mentioned experiments and Acronym Overlap and in the modified version. I opt to accept this paper.

---

> > > ### Author Response · Authors · 2023-08-18
> > >
> > > Thanks for your time and valuable comments, we will keep on this direction of research.
> > >
> > > Best wishes.

---

### Official Review · Reviewer_cAj4 · 2023-07-06

**Soundness:** 3 good
**Presentation:** 3 good
**Contribution:** 3 good
**Rating:** 6
**Confidence:** 3

**Summary:**

The paper proposes a variational structured memory module (VSM) that can be combined with both VAE and Diffusion type models to improve their performance on few-shot generation tasks. The memory structure is both hierarchical and structured into episodic and semantic memory components such that the model can learn to retain detailed experiences as well as higher level general information useful for the task.

**Strengths:**

- VSM is technically sound and builds on a hierarchical conditioning of latent variables and an attention-structure to retrieve information at inference time.
- The experiments show that VSM can be combined with existing few-shot generation models (both VAE and Diffusion type models) to improve their performance making it a modular addition widely applicable.
- An ablation study shows efficacy of the memory types and update strategy

**Weaknesses:**

- The computational overhead of applying VSM is not discussed. Especially VSM-VAE contains many autoregressive conditionals that would have to be evaluated in sequence. How does the inference runtime compare to not applying VSM?
- How sensitive is the optimization procedure of VSM to hyperparameters in comparison to not using VSM? Intuitively, the addition of ELBO terms for the memory components could make optimization trickier. Is this observed in practice? How do you deal with this added complexity?

**Questions:**

- It would be interesting to see whether the approach also scales to more challenging datasets such as ImageNet. Are the limitations purely computational or would you expect other challenges?
- Why is a top-K retrieval strategy used when attention could be applied over all available elements instead? Based on the hyperparameters in the supplementary, the size of the memory (even including episodic memory) does not seem prohibitively large. How would the results change without retrieval, or as you change K?

**Limitations:**

- How well does the method scale to larger model sizes and/or dataset sizes?
- The influence of the number of few-shot examples is not discussed. What are the challenges as the number of examples decreases/increases?

---

> ### Author Rebuttal · Authors · 2023-08-10
>
> Thanks for your constructive comments and suggestions, which are helpful for us to further improve the quality of our paper. The concerns have been addressed as below.
>
> **W1**: Concern on computational overhead of applying VSM.
>
> **A1**:  Thanks for your constructive suggestion. We have provided the additional inference time caused by introducing VSM in the following table. From the results, we can find that the developed VSM-based methods will add no more than 10$\%$ to the inference time compared to these baselines without VSM, which is quite slight in our understanding.
>
> | Models | Inference time(s)/(Batch_size=32)|  |
> | ----- | -------|  ------- |
> |CNS| | 2.77| 0 |
> |CNS + Semantic | 2.81 | +0.14 |
> |CNS + Episodic | 3.37 | +0.60|
> |VSM-CNS | 3.0924| + 0.32 |
>
> | Models | Inference time(s)/(Batch_size=32)| Rate |
> | ----- | -------|  ------- |
> |SCHA-VAE | 4.17| 0 |
> |SCHA + Semantic | 4.32 | +0.15 |
> |SCHA + Episodic | 6.47 | +2.30 |
> |VSM-SCHA | 4.50| + 0.33  |
>
> | Models | Inference time(s)/(Batch_size=32)| Rate |
> | ----- | -------|  ------- |
> |sDDPM | 98.52 | 0|
> |vDDPM |  92.23 | -5.67|
> |vFSDM | 103.98 | +5.46|
> |VSM-Diffusion | 107.59 | +9.07|
>
> **W2**: Concern hyperparameters in comparison to not using VSM? The addition of ELBO terms for the memory components could make optimization trickier. Is this observed in practice? How do you deal with this added complexity?
>
> **A2**: The optimization of VSM-based models is stable and doesn't face any challenges. Actually, the selection of hyperparameters won't heavily influence the performance of our method. For the hyperparameter $\alpha$ in the mechanism of memory update, we have conducted the experiment to elaborate on the impact of $\alpha$, which won't influence the model performance too much as shown in the following table.
>
> |Model| $\alpha$ | Onmignot| | Double-mnist| |Mnist | |
> |------|-------|------|-------|------|-------|------|------|
> | | | ELBO | NLL | ELBO | NLL| ELBO| NLL|
> |VSM-CNS |0.1 |83.1| 69.5| 49.3| 42.7| 93.2| 77.6|
> |VSM-CNS | 0.5| 83.2 |69.9 |48.1 |42.3 |93.5 |77.8|
> |VSM-CNS |0.7| 81.1 |68.2| 47.8| 40.6| 92.9| 77.0|
> |VSM-CNS |0.9 |84.1 |70.8 |48.7 |42.6 |93.9 |78.5|
> |VSM-SCHA | 0.1 |74.6 |57.0 |42.4 |35.6 |89.7 |68.1|
> |VSM-SCHA |0.5| 73.4| 56.2| 42.2| 35.3| 88.4 |67.1|
> |VSM-SCHA |0.7 |74.0| 56.8 |42.4| 36.5| 89.9 |68.2|
> |VSM-SCHA |0.9 |74.8| 57.5| 43.8| 37.9| 90.1 |69.7|
>
> For the other hyperparameters in generative models, we directly follow the default hyperparameter settings of baselines to make a fair comparison.
>
> **Q1**: Scales to more challenging datasets such as ImageNet.
>
> **A1**: For scaling our methods to more challenging datasets such as ImageNet, in our undstanding, the computation cost would be the only challenge. As you can see, it will be extremely time-consuming to evaluate our method on the dataset in such scale. Moreover, to improve the generative performance, we may need to increase the number of blocks in the developed structured memory module.
> **We have provided some initial results on miniimagenet in our response to all reviwers** and will try to include more results on imagenet in the revision.
>
> **Q2**: Why is a top-K retrieval strategy used when attention could be applied over all available elements instead? ... How would the results change without retrieval?
>
> **A2**: Actually, the top-K retrieval strategy can be seen as an approach to making the attention weight more sparse, which is effective at alleviating overfitting in attention models [3].  Then, top-K retrieval strategies [5] can also speed up the training, distilling noises without losing information, and increasing the model performance.  Indeed, top-K retrieval strategies have been wildly applied in various memory-augment models [1, 2].
>
> To evaluate the effectiveness of top-K retrieval strategy, we provide additional experimental results with various settings of K on Omniglot dataset.
>
> |top-K proportion | Onmignot|  | Double-mnist| |Mnist |   | Training time (mini-batch)|
> | ----- | -------|  ------- |   ----- | -------|  ------- |  --------- | --------- |
> |-- | ELBO|NLL| ELBO|NLL| ELBO|NLL| -- |
> |1%| 73.8| 55.4| 43.2| 36.8| 89.3| 68.5| 79.3s|
> |50%| 77.2| 59.9| 46.8 |39.6| 92.2| 73.8| 86.4s|
> |100% |78.3 |59.6| 47.9 |41.5| 92.2 |76.7 |95.2s|
>
> [1] Zhen X, Du Y, Xiong H, et al. Learning to learn variational semantic memory[J]. Advances in Neural Information Processing Systems, 2020, 33: 9122-9134.
>
> [2] Blattmann A, Rombach R, Oktay K, et al. Retrieval-augmented diffusion models[J]. Advances in Neural Information Processing Systems, 2022, 35: 15309-15324.
>
> [3] Fan, Xinjie, et al. "Bayesian attention modules." Advances in Neural Information Processing Systems 33 (2020): 16362-16376.
>
> [4] Wang, Pichao, et al. "Kvt: k-nn attention for boosting vision transformers." European conference on computer vision
>
> **L1**:How well does the method scale to larger model sizes and/or dataset sizes?
>
> **A1**: We have discussed our method scale to larger dataset sizes in the response to Q1. As for larger model sizes, we believe that the high-level idea of our method can be flexibly applied on large-scale model, like Stable Diffusion, but there will be a lot of technical details needed to be considered in the process of implementation. We will try to explore this direction to introduce memory module into large-scale foundation models.
>
> **L2**: What are the challenges as the number of examples decreases/increases?
>
> **A6**: Actually, there exists a previous work [5] that has discussed the influence of the number of examples decreases/increases to few-show generation. Generally speaking, providing more data samples of a new task can help the generative model to understand the statistical properties of dataset more comprehensively, and further improve the performance of few-show generation.
>
> [5] Giannone G, Winther O. Scha-vae: Hierarchical context aggregation for few-shot generation[C]//International Conference on Machine Learning. PMLR, 2022: 7550-7569.

---

> > ### Comment · Reviewer_cAj4 · 2023-08-16
> >
> > I would like to thank the authors for clarifying my questions and adding additional experiments that support the claims and contributions of the paper.
> >
> > I encourage the authors to incorporate the additional insights into the paper and supplementary material. My recommendation remains to accept the paper.

---

> > > ### Author Response · Authors · 2023-08-18
> > >
> > > Thanks for your time and valuable comments.
> > >
> > > Best wishes.

---

### Official Review · Reviewer_xiFB · 2023-07-07

**Soundness:** 3 good
**Presentation:** 2 fair
**Contribution:** 3 good
**Rating:** 6
**Confidence:** 2

**Summary:**

This work is focused on the problem of few-shot generation using a memory mechanism. A few priors works have leveraged semantic memory or short-term (episodic) memory for such tasks. In this work, a new architecture is proposed named Variational Structured Memory (VSM) that leverages both types of memory mechanisms. Two variants are proposed: VSM-VAE and VSM-Diffusion. The paper discusses the training objective used to train the model and also present ways for recall (using attention and kNNs) from the memory when a new task is available. Lastly, the paper discusses how the memory is updated and initialized. Experiments conducted on existing datasets show that the method outperforms the baselines. Furthermore, the paper also reports the generated images and study the effect of memory size and the memory update process.

**Strengths:**

1. The proposed idea of using both types of memory for recall is interesting.
2. The experiments are thorough and support the claims made in the paper.
3. The paper is well-written and easy to follow.


**Weaknesses:**

1. Some design choices of the algorithm can be explained better (more on this in Questions below).
2. Limitations of the current method are not discussed. The discussion should include what can be other strategies for updating semantic and episodic memories. The paper should also talk about how to use such an architecture for generating images that can be out of distribution for the new dataset.


**Questions:**

1. Semantic Memory Update: The target for the mean of the semantic memory vector seems to favor examples that are close to the current vector of semantic memory using the g(.) function. Why not simply just move the semantic memory towards the mean of all values in vector $H_n$?
2. Adding to the previous question, the update will favor examples that are closer to the current value of the semantic memory. Will this lead to a mode collapse where the model is unable to generate diverse samples for a category? Is there a way to validate if the model has diversity in the generated samples?
3. From my understanding, the number of semantic memory modules is the number of categories (N). What if we need to have a different size of semantic memory than the category? Will the model still work?
4. Will the method work for out-of-distribution datasets during evaluation because the memories are updated by having higher weights for examples closer to the current memory?
5. At line 113, should Eq. 5 be referred to instead of Eq. 9?
6. Minor Typos:
 - Line 135- space between ‘themechanism’
 - Line 151- ‘distribution’ spelling is incorrect



**Limitations:**

Limitations are not discussed in the paper.

---

> ### Author Rebuttal · Authors · 2023-08-10
>
> We appreciate your constructive comments and suggestions, which are helpful for us to further improve the quality of our paper. The concerns have been addressed as below.
>
> **W1**: Some design choices of the algorithm can be explained better.
>
> **A1**: Thanks, we will explain your mentioned questions one by one in the following response.
>
> **W2**: Limitations of the current method are not discussed.
>
> **A2**: Thanks for your insightful suggestions. We have discussed other strategies for updating semantic and episodic memories in the response to Q1, and claimed the setting of OOD generation in the response to Q4. We will included these discussions in our revision.
>
> **Q1**: Semantic Memory Update: The target for the mean of the semantic memory vector seems to favor examples...... Why not simply just move the semantic memory towards the mean of all values in vector?
>
> **A1**: Thanks for your valuable suggestions. Actually, we used to consider to utilize the average of the vectors stored in episodic memory to update the semantic memory, but we assume that introducing attention mechanism can highlight the contributions of typical data samples for better model performance. For instance, a typical data sample can serve as the semantic memory of a specific category in the corresponding memory block. To verify our thoughts, we conduct the following experiments to compare these two memory update methods, and the experiment results shown in the table demonstrate the superior of the attention mechanism. Similar conclusions can be found in [1].
>
> | | omniglot | |double-minst ||mnist |   |
> | ----- | -------|  ------- | --------- | --------- |   --------- |  --------- |
> |Update method |  ELBO| NLL| ELBO | NLL | ELBO| NLL |
> | Mean |      77.7 | 60.1 | 46.0| 39.5| 93.8| 70.3|
> | Ours |   74.0 |56.8| 42.4| 36.5| 89.9| 68.2|
>
> **Q2**: the update will favor examples that are closer to the current value of the semantic memory. Will this lead to a mode collapse where the model is unable to generate diverse samples for a category? Is there a way to validate if the model has diversity in the generated samples?
>
> **A2**: We note that the latent representations of data samples retrieved from our memory module only serves as a prior for the downstream generation task, and wouldn't lead to the phenomena of **mode collapse**.
> For the qualitative results shown in Fig.3, for a novel unseen generation task, we can find that the visualized data samples generated by our model are diverse. Moreover, the metric precision (p) in our experiments can be used to measure the performance of diversity [2]. From the results in the following table, we can find that our developed VSM-based models can outperform on the metirc of precision than other baselines.
>
> | Update method| FID  |SFID | P |  R |
> | ----- | -------|  ------- | --------- | --------- |
> | Base |    13.52| 27.75| 0.71| 0.38|
> | Mean |    12.33| 25.82| 0.72| 0.40|
> | GAT |   12.27 |25.79| 0.72| 0.40|
>
> [1] Zhen X, Du Y, Xiong H, et al. Learning to learn variational semantic memory[J]. Advances in Neural Information Processing Systems, 2020, 33: 9122-9134.
>
> [2] Tuomas Kynkäänniemi, Tero Karras, Samuli Laine, Jaakko Lehtinen, and Timo Aila. Improved precision and recall metric for assessing generative models. Advances in Neural Information Processing Systems, 32, 2019.
>
> **Q3**: What if we need to have a different size of semantic memory than the category? Will the model still work?
>
> **A3**: It's an interesting question. Actually, each vector in the semantic memory module can be treated as the clustering center of the corresponding memory block, and the optimal setting of the clustering number should be the number of categories (N), which is also used in [1].  However, it should also work for these memory-augmented models to set less memory blocks, where some similar categories can be summarized with a single memory block. We conduct the following experiment to verify our thoughts as shown in the following table, and the experimental results demonstrate that our model can still work by setting the semantic memory size smaller than the number of categories.
>
> |proportions|Omniglot||Double-mnist||Mnist||
> | ----- | -------|  ------- | --------- | --------- | -------|  ------- |
> | |ELBO| NLL |ELBO| NLL| ELBO| NLL|
> |SCHA + Episodic| 82.5| 63.4| 53| 47.2| 95.3| 79.1|
> |0.1% |81.8| 62.7 |52.2| 46.9 |95.1| 78.6|
> |10% |78.7 |61.5 |46.0 |39.3 |94.6| 72.5|
> |30% |78.6 |60.2 |46.1 |40.2 |94.2 |71.8|
> |50% |76.4| 60.8 |46.0 |39.7 |92.3| 71.9|
> |70% |77.0 |60.2 |44.5 |39.4 |92.2 |71.6|
> |90% |74.6 |57.5| 43.3 |36.4| 90.7 |69.1|
> |100% |74.0 |56.8|42.4 |36.5| 89.9 |68.2|
>
> **Q4**: Will the method work for out-of-distribution datasets during evaluation because the memories are updated with higher weights for examples closer to the current memory?
>
> **A4**: Thanks for your question. Actually, in the setting of few-shot generation, the evaluation is conducted on out-of-distribution datasets, which has been described in Section 3.1. Generally speaking, the generative models will be firstly trained with a series of old tasks (data samples of those seen classes) in the training stage and tested on new tasks (data samples from the other unseen classes). For example, the data samples of old tasks consist of birds, tigers, leopards, and elephants, while the data samples of new tasks will include wolves, dogs and other unseen animals. Thus, in the setting of few-shot generation, the model can learn how to generate OOD data samples by utilizing data information stored in old tasks.
>
> **Q5**: At line 113, should Eq. 5 be referred to instead of Eq. 9?
>
> **A5**. Yes, thanks for your careful verify. We will delete Eq.9 in our revised manuscript.
>
> **Q6** Minor Typos:
> Line 135-  ‘themechanism’
> Line 151- ‘distribution’
>
> **A6**: Thanks. We will fixed these typos in our revision.
>
> Thanks again for your valuable advices on improving our paper, and we hope to take more discussions with you to improve this work.

---

> > ### Comment · Reviewer_xiFB · 2023-08-17
> >
> > I thank the authors for the detailed clarification of my doubts.
> >
> > I have a doubt regarding A3- Were there any major changes to the architecture when the number of memory blocks was less than the number of categories?

---

> > > ### Author Response · Authors · 2023-08-18
> > >
> > > Thanks for your further discussion. After reducing the number of memory blocks, which will be less than the number of categories, there should be another metirc to map each training sample to its corresponding memory block. In our case, we simply adopt the idea of clustering, and assign each training sample to its nearest clustering center, where the distance between the latent representation and the semantic vector of its assigned memory block should be closer than the others.
> > >
> > > As for the network architecture, to fairly conduct an ablation study on the impact of the number of memory blocks, we did not modify the main  body of VSM-SCHA, and only reduced the corresponding amount of memory blocks, whose proportion accounts for 0.1%-90% of the memory number of our original model.  As the results shown in A3, although the model performance becomes worser with the reduction of the number of memory blocks, our method can still outperform the baseline SCHA+VAE shown in Table.1 of our paper, even under the case with only one memory block, specificallly 0.1%.
> > >
> > >
> > >
> > >
> > >
> > >
> > >
> > > Although the performance  with the reduction of memory block, our memory mothod still works

---

> > > > ### Comment · Reviewer_xiFB · 2023-08-18
> > > >
> > > > Thank you for the detailed description. I hope the authors update the paper with these experiments and I have updated my score.

---

> > > > > ### Author Response · Authors · 2023-08-18
> > > > >
> > > > > Thanks for your time and valuable comments. We will included these experiments into our revision as suggested by you.
> > > > >
> > > > > Best regards.

---

### Official Review · Reviewer_zaLT · 2023-07-26

**Soundness:** 3 good
**Presentation:** 2 fair
**Contribution:** 3 good
**Rating:** 6
**Confidence:** 3

**Summary:**

The authors propose to incorporate 'episodic' and 'semantic' memory inspired by the human memory system into existing generative modeling frameworks to improve few-shot generation abilities. Specifically, they propose a variational structured memory module (VSM) and show its incorporation into existing VAE based models and diffusion based models can improve few-shot generation across multiple datasets. The memory mechanism is introduced as a prior to the context variables in traditional neural statistician (NS) and diffusion modeling frameworks. The ‘semantic’ memory is modeled as a ‘general’, lightweight storage with quicker retrieval (single embedding for each category) while ‘episodic’ memory is modeled to be more detailed/'vivid' (a set of embeddings for each category) and provide context relevant information. Learning involves an attention-based recall/lookup from prior memory states and subsequent update of the semantic and episodic memory for each category.

**Strengths:**

1. The integration and formulation of proposed memory modules (and relevant storage, recall and update mechanisms) to improve few-shot generation is relatively novel.
2. The experiments are performed on established datasets and the results support the claim that the proposed VSM module can benefit performance on few-shot generative/diffusion-based tasks.
3. The method and underlying components are largely adequately described in section 3 and 4 with relevant references or background provided.

**Weaknesses:**

1. The work's claim of modelling the memory mechanism like a 'human being' for few-shot generation tasks seems far-fetched and not adequately backed by literature or explanation. While episodic and semantic memory indeed are components of human memory (and some references for this are provided), it is not clear how they are related to static image few-shot generation (specifically, episodic memory is largely related to an agent's experiences/events and seems more relevant to episode/reinforcement learning based tasks/frameworks).

2. Further, it remains unclear what is the authors mean when they mention 'context' in relation to the proposed semantic and episodic memory. E.g. on L146, it is mentioned 'semantic memory in human brain can provide context information'; while L139 mentions 'episodic memory can provide context information'. So it is unclear if both semantic and episodic are to provide 'context' and if so what does 'context' mean here? Again, in relation to my first point, backing statements such as L146 with references would be beneficial.

3. Experiment setup: It is unclear whether all models were evaluated with the same vision encoder and processing. (In appendix L401 authors mention they draw inspiration from vision transformer in processing image features and suggest their process differs from existing memory-based models). Were other models evaluated in a similar setup for fair comparison? Otherwise, performance benefits might be due such differences in experimental setup and not necessarily the proposed memory method itself.

4. Further, authors should consider indicating the parameters of their method and other relevant factors to provide a better comparison to existing models (e.g. see Table1 in SCHA-VAE: Hierarchical Context Aggregation for Few-Shot Generation Giorgio Giannone, Ole Winther International Conference on Machine Learning, ICML, 2022). It is currently unclear whether the performance benefits may merely be due to more parameters or if it is due to the proposed method.

5. While source code is provided in appendix, would it be released publicly for reproducibility? (currently there is no indication). Also, how many trials were performed for each experiment?

Minor:
1. The writing in introduction has minor grammar mistakes and missing words / wrong phrasing (e.g. L9: 'to assistant' -> 'to assist'; L18: 'that never encountered before' -> 'that it has never encountered before'; L24: 'which mainly' -> 'which is mainly'; L49 'to assistant' -> to assist', etc)




**Questions:**

Please see weaknesses section above.

**Limitations:**

A brief limitation section is provided in the appendix L449. Perhaps it could be expanded to discuss potential negative applications of generative/diffusion-based methods.

---

> ### Author Rebuttal · Authors · 2023-08-10
>
> Thanks for your constructive comments and suggestions.
>
> Q1.  VSM for static image few-shot generation.
>
> A1: Thanks for your valuable question. Actually, the motivation of our work is to develop a series of memory-augmented generation models to imitate the creative process of humans. For instance, when a painter is required to paint an image with a bird, he will first recall the conceptual information (semantic memory) of a bird equipped with a series of realistic bird samples (episodic memory), and then fuses these two kinds of memory information to accomplish this creative task. And this is also the reason why we want to imitate the memory mechanism of human to redefine the generative process of existing generative models. Similar idea of introducing memory mechanism can be found in other works [1,2,3,4].
>
> Moreover, memory modules have also been widely used in few-shot learning tasks. For example,  LSTMs trained to meta-learn can quickly learn never-before-seen quadratic functions with a low number of data samples[5]. And [6] builds an effective memory augment meta-learning framework for few-shot learning with an attention module.Recently, [7] proposed variational semantic memory and applied it to the few-shot classification task, where the intuition behind this is that the model can utilize semantic information (semantic memory) gained from past tasks to solve a new task. These two mentioned aspects inspire us to develop a variational structured memory module and apply it on few-show generation task.
>
> As for the episodes in RL, in our understanding, the episodes of RL, which will be stored in buffer, are only used to update the policy network, but won't be recalled during testing. But we believe that the idea of introducing memory module can be also extended for improving the performance of existing RL-based methods.
>
> Q2. it is unclear if both semantic and episodic are to provide 'context' ......?
>
> A2. We're sorry for the confusion caused by our unclear description. Indeed, there are two advanced forms of memory in the human brain, specifically semantic memory allows the storage of general conceptual information [7] and episodic memory allows the collection of detailed episodes [8]. We have highlighted these descriptions in the main paper and modified the sentence ``episodic memory can provide context information that will later be store in semantic memory''  `` into ``episodic memory can provide detailed episodes that will be converted into conceptual information and stored in semantic memory``. We have also modified other unclear descriptions about context information in our paper.
>
> Q3. Fair comparison?
>
> A3. Sorry for the misunderstanding. Actually, the only difference between our method and other baselines is the introduction of the proposed variational structure memory module and the remained factors, like network structures, experimental settings and evaluation metrics,
> are exactly the same as described in their papers. Specifically, we introduce the variational structure memory module into VAE-based models [9] and diffusion models [10], leading to VSM-VAE and VSM-diffusion, respectively. And we directly follow and implement our method on their released code to make a fair comparison, which can be checked in our submitted supplemental material. We have also listed the network structures in comparsion as follows:
>
> | Model |  Encoder|
> | -------- | --------- |
> |CNS |ResNet|
> |SCHA | ResNet|
> |VSM-CNS |ResNet|
> |VSM-SCHA | ResNet|
> |sDDPM | ViT|
> |vDDPM | Unet|
> |vFSDM  | ViT|
> |VSM-Diffusion | ViT|
>
> The description in Appendix L401 is only used to illustrate the implementation details of variational structure memory module, which can be treated as another slight contribution of our method to existing memory-augment models and will not cause unfairness in experimental comparisons.
>
> Q4.  More parameters.
>
> A4. Thanks for your valuable advice. Actually, as discussed in Q3, the main body (encoder and decoder) of the network structures used in VSM-VAE and VSM-Diffusion are still the same as their corresponding baselines, and the only additional cost of parameters is brought by introducing the proposed variational structured memory module, which will not bring too much additional memory cost.
> To make a further investigation, we list the amount of parameters of whole generative models and their memory modules in the following table, and we can find that the proposed variational structure memory will not bring too much increase in the number of parameters.
>
> | Models | Parameters (M)| Rate |
> | ----- | -------|  ------- |
> | CNS | 7.34|  0%|
> | CNS + semantic| 7.39|  0.68% |
> | CNS + Episodic | 7.38|  0.54%|
> | VSM-CNS | 7.44| 1.36% |
>
> | Models | Parameters (M)| Rate |
> | ----- | -------|  ------- |
> | SCHA | 5.37|  0% |
> | SCHA + semantic| 5.60|  +4.28% |
> | SCHA + Episodic | 5.55|  +3.35% |
> | VSM-SCHA| 5.77|  +7.44% |
>
> | Models | Parameters (M)| Rate |
> | ----- | -------|  ------- |
> | sDDPM| 34.0|  0% |
> | vDDPM| 35.5|  4.41%|
> | vFSDM | 34.8|  2.35%|
> | VSM-Diffusion| 35.6|  4.70% |
>
> Q5. Source code, how many trials......?
>
> A5. Thanks for your notification. As shown in the supplemental material, we have included the shell files to reproduce our results released on the paper. And we promise that we will absolutely release a more formal version of source code on the github, equipped with a readme file to guide the reproducibility of our work. To make a fair comparison, for each experiment, we take five trials and report the average of these obtained results.
>
> [1] Variational memory addressing in generative models.[2]Variational memory encoder-decoder.[3] Memory transformer.
> [4] Recurrent memory transformer.[5]Learning to learn using gradient descent.[6] Meta-learning with memory-augmented neural networks
> [7]Learning to learn variational semantic memory.[8] Retrieval-augmented diffusion models.[9] Scha-vae: Hierarchical context aggregation for a few-shot generation.[10] Few-shot diffusion models.

---

> > ### Comment · Reviewer_zaLT · 2023-08-15
> >
> > Thanks for the detailed response and clarifications.
> >
> > Regarding point 1, perhaps the distinction between semantic and episodic memory in the specific context of few-shot learning can be better described in the main paper as currently done in rebuttal.
> >
> > Regarding point 4, the parameter cost could be included in the original table to highlight the performance benefits of the method with relatively low additional parameters. Point 3 could be mentioned in supplemental to inform the reader.
> >
> > As concerns have been addressed, I've raised score to 6.

---

> > > ### Author Response · Authors · 2023-08-15
> > >
> > > Thanks for your time and valuable comments.
> > >
> > > Best regards.

---

### Author Rebuttal · Authors · 2023-08-10

We really appreciate all the reviewers for their constructive and helpful comments, which can greatly help us to improve the quality of our paper. For the mentioned typos, grammar mistakes, unclear notations and missing citations, we promise that we will carefully fix them  in our revised paper. For all reviewers' questions and suggestions, we have responded one by one in the blow, and you can find it in the corresponding block.



Due to the page limitation, we put a small part of the supplementary experimental results of few-show generation that training the generative models on FScifar100 and then testing them on MiniImageNet in this block.


| model              | **FID**   | SFID      | P        | R        |
| ------------------ | --------- | --------- | -------- | -------- |
| DDPM               | 63.13     | 33.23     | 0.61     | 0.30     |
| sDDPM              | 47.73     | 32.90     | 0.56     | 0.37     |
| vFSDM              | 61.57     | 32.65     | 0.59     | 0.31     |
| FSDM               | 42.32     | 25.74     | 0.61     | **0.37** |
| VSM-Diffusion(our) | **38.57** | **24.80** | **0.65** | 0.36     |

---

### Comment · Area_Chair_ZH2c · 2023-08-14
**engage in the discussion and raise any questions about the rebuttal, if any**

Dear reviewers,

The authors have submitted a rebuttal. Please go through the rebuttal and other reviewers' reviews. Feel free to clarify your doubts and raise questions to the authors, if any.

Thank you

---

### Decision · Program_Chairs · 2023-09-21

**Decision:**

Accept (poster)

**Comment:**

All reviewers reached a consensus to accept the paper. The authors are encouraged to incorporate all the feedback and suggestions from the reviewers into the final version of the manuscript. Moreover, the authors are also strongly encouraged to benchmark the methods on more complex datasets, such as mini-ImageNet.

After internal discussions, we also recommended the authors to revise the title. "like human being" seems to be a big claim as it lacks solid evidence to support this point explicitly in the paper. Changing to "brain-inspired" or some sort would be more appropriate.

**An important note from the SAC:**
We value the technical strength of the paper, but have some critical concerns, which we ask the authors to address. We decided not to recommend to reject the paper because we believe the authors will respect our note, and follow the changes we ask. We hope our trust is justified. Beyond issues raised by the reviewers and AC in previous discussions, the following issues came up in discussion recently and are critical (but also easy) to address:

1. The title makes an unjustified claim that the approach follows the cognitive process of "human beings". This is not substantiated with any finding from neuroscience/cogsci. As far as we know, this is likely patently incorrect. We ask the authors to change the title and remove any reference to human processes from the paper. It's OK to say in the discussion that you are inspired by human reasoning, while providing references from neuroscience/cogsci. But the claim currently reflected in the title is simply misleading.

2. We ask the authors to remove any reference to their approach as "bionic", as it's currently described in the abstract. Bionic is defined as:
> having artificial body parts, especially electromechanical ones.
This does not apply to the approach. Please remove this description and any similar description from the paper.

Especially in the current days of AI hype, it's critical not to make such claims without the strongest proof possible. It will be irresponsible to see such claim at NeurIPS, one of our field's top venues.

Thanks,

The SAC